    

# Macropinocytosis mediates resistance to loss of glutamine transport in triple-negative breast cancer

Kanu Wahi [1,8✉], Natasha Freidman[1,2,3,8], Qian Wang[1,4], Michelle Devadason[1,5], Lake-Ee Quek[6], Angel Pang[1], Larissa Lloyd[1], Mark Larance[2], Fabio Zanini [5], Kate Harvey[7], Sandra O'Toole[2], Yi Fang Guan[1] & Jeff Holst [1✉]

## Abstract

**Triple-negative breast cancer (TNBC) metabolism and cell growth uniquely rely on glutamine uptake by the transporter ASCT2. Despite previous data reporting cell growth inhibition after ASCT2 knockdown, we here show that ASCT2 CRISPR knockout is tolerated by TNBC cell lines. Despite the loss of a glutamine transporter and low rate of glutamine uptake, intracellular glutamine steady-state levels were increased in ASCT2 knockout compared to control cells. Proteomics analysis revealed upregulation of macropinocytosis, reduction in glutamine efflux and increased glutamine synthesis in ASCT2 knockout cells. Deletion of ASCT2 in the TNBC cell line HCC1806 induced a strong increase in macropinocytosis across five ASCT2 knockout clones, compared to a modest increase in ASCT2 knockdown. In contrast, ASCT2 knockout impaired cell proliferation in the non-macropinocytic HCC1569 breast cancer cells. These data identify macropinocytosis as a critical secondary glutamine acquisition pathway in TNBC and a novel resistance mechanism to strategies targeting glutamine uptake alone. Despite this adaptation, TNBC cells continue to rely on glutamine metabolism for their growth, providing a rationale for targeting of more downstream glutamine metabolism components.**

**Keywords** ASCT2; Glutamine Metabolism; Macropinocytosis; Triple-Negative Breast Cancer; Metabolomics
**Subject Categories** Cancer; Metabolism

## Introduction

Metabolic reprogramming is essential to support the high energy demand of proliferating cancer cells (Hanahan and Weinberg, 2011; Hanahan, 2022). Glutamine addiction is an example of altered metabolism wherein some cancer cells rely on glutamine, a non-essential amino acid in normal cells, to sustain proliferation (Wei et al, 2021b). This increased reliance on glutamine is evident in some aggressive cancer subtypes, such as triple-negative breast cancer (TNBC), that currently lack good therapeutic options (Winnike et al, 2018; Van Geldermalsen et al, 2016; Quek et al, 2022). Hence, there has been a keen interest in understanding glutamine metabolism as a potential therapeutic target in TNBC (Jeon et al, 2015; Van Geldermalsen et al, 2016; Timmerman et al, 2013; Kung et al, 2011).

Knockdown of the glutamine transporter ASCT2/SLC1A5 can dramatically reduce proliferation in vitro and tumour growth in xenograft models of TNBC (Jeon et al, 2015; Van Geldermalsen et al, 2016). On the contrary, the proliferation of the TNBC cell line, HCC1806, is unaffected by ASCT2 KO, despite the activation of amino acid stress response pathways (Bröer et al, 2019). These data suggest that complete loss of ASCT2, as opposed to partial loss, triggers compensatory mechanisms that facilitate growth adaptation to impaired glutamine uptake. Since ASCT2 has been suggested as a therapeutic target across a range of cancers, with multiple studies attempting to target ASCT2 (Jin et al, 2023), it is critical to determine if there are key resistance mechanisms to targeting ASCT2.

One potential resistance pathway is macropinocytosis, a nutrient scavenging mechanism observed in some tumour cells wherein extracellular nutrients such as amino acids, proteins and lipids are engulfed from the microenvironment and used to support cellular metabolic processes. Previous work has shown that some cell lines exhibit constitutive macropinocytosis, while in other cell lines, macropinocytosis is induced when nutrients are restricted or in response to cytotoxic drug treatments (Lee et al, 2019; Jayashankar and Edinger, 2020). AMPK is known to drive the activation of AKT under nutrient stress, both of which are upstream effectors of macropinosome formation (Kim et al, 2018; Han et al, 2018). This also involves the recruitment of sorting nexin proteins (SNX1, 5, 9 and 33), Rac1 and Pak1 to the macropinosomes (Finicle et al, 2018; Zhang and Commisso, 2019; Lim and Gleeson, 2011).

In this study, we set out to explore the growth adaptation observed upon loss of ASCT2, expanding the HCC1806 data across four additional breast cancer cell lines, including both TNBC (glutamine addicted), Luminal A (glutamine independent) and

[1]School of Biomedical Sciences, UNSW Sydney, Kensington, NSW, Australia. [2]School of Medical Sciences, The University of Sydney, Sydney, NSW, Australia. [3]Cell Biology Program, Memorial Sloan Kettering Cancer Center, New York, NY, USA. [4]Children's Cancer Institute, Lowy Cancer Research Centre, UNSW Sydney, Kensington, NSW, Australia. [5]School of Clinical Medicine, UNSW Sydney, Kensington, NSW, Australia. [6]School of Mathematics and Statistics, University of Sydney, Sydney, NSW, Australia. [7]Cancer Ecosystems Program, Garvan Institute of Medical Research, UNSW Sydney, Kensington, NSW, Australia. [8]These authors contributed equally: Kanu Wahi, Natasha Freidman. ✉E-mail: k.wahi@unsw.edu.au; j.holst@unsw.edu.au

HER2[+] breast cancer cell lines. Our data show distinct metabolic adaptation to ASCT2 knockout within the TNBC subtype, including upregulation of macropinocytosis. These data suggest a cautious approach to targeting glutamine uptake through ASCT2, which, like glutaminase inhibition, has a number of potential resistance pathways.

# Results

## ASCT2 protein is highly expressed in TNBC patient samples

We have previously shown high *SLC1A5* (ASCT2) mRNA expression using Nanostring analysis in a cohort of 96 TNBC patient samples (Van Geldermalsen et al, 2016). To determine whether the high TNBC mRNA levels correspond to ASCT2 membrane protein expression, we undertook IHC analysis and scoring of an expanded cohort of 155 TNBC samples (Fig. 1A,B). ASCT2 protein was expressed on the membrane in 154 out of 155 TNBC samples, with the majority exhibiting maximal membrane expression at score 2 or score 3 (Fig. 1B) and H-score analysis showed a spread of membrane expression levels across the TMA (Fig. 1C). Interestingly, ASCT2 membrane expression (max. H-score) showed a significant correlation with PD-L1 expression in both epithelial ($p = 0.0032$) and stromal ($p = 0.0141$) cells, consistent with a previous report (Ansari et al, 2020).

## Complete loss of ASCT2 triggers a growth adaptation in TNBC cells

We first set out to determine the impact of complete loss of ASCT2, using CRISPR/Cas9 to target exon 7 of the *SLC1A5* (ASCT2) gene in HCC1806 cells (Fig. 1D; Appendix Fig. S1). We screened several clonal lines derived from transfected single cells for the absence of ASCT2 protein expression and selected six separate clones for further analysis (Fig. 1E; K1–K6). Since standard tissue culture media has non-physiological levels of glutamine and other nutrients, we performed the ASCT2 CRISPR/Cas9 knockout and cloning steps in standard 2 mM glutamine (K2 and K5), or at physiological 0.5 mM glutamine (K1, K3, K4, K6). A significant reduction in glutamine uptake for all six ASCT2 knockout lines was observed using radiolabelled glutamine uptake assays, confirming the functional loss of the ASCT2 glutamine transporter (Fig. 1F). However, despite impaired glutamine transport, the ASCT2 knockout cells showed no difference in cell viability in comparison to the WT cells (Fig. 1G). This was in stark contrast to published siRNA and shRNA phenotypes for ASCT2 knockdown in HCC1806 and other TNBC cell lines (Jeon et al, 2015; Van Geldermalsen et al, 2016). Due to the potential impact of unphysiological in vitro glutamine concentrations on growth, we further examined the ASCT2 KO K2 clone using an orthotopic xenograft mouse model in vivo. Comparison of the HCC1806 WT and K2 xenografts showed no significant difference in the tumour weights (endpoint) or tumour volume over 3 weeks prior to the ethical endpoint (Fig. 1H,I). Again, this was inconsistent with previous siRNA and shRNA in vivo phenotypes (Jeon et al, 2015; Van Geldermalsen et al, 2016), and confirmed that the lack of a

growth phenotype was not due to an in vitro artefact of high glutamine concentrations.

To address the potential effects of the CRISPR/Cas9 procedure, we transfected the HCC1806 cells with either a CRISPR/Cas9 non-targeted control (CR-NC) plasmid or our ASCT2 targeted plasmid (CR-A2). Additionally, to eliminate the confounding effect of expanding ASCT2 knockout cells from a single-cell clone, we used fluorescence-activated cell sorting to purify a polyclonal population of transfected cells using an ASCT2 antibody that can reversibly bind to the extracellular domain of ASCT2 (Fig. 1J). Post-sorting analysis of the polyclonal HCC1806 A2KO cells showed successful knockout by both flow cytometry (Fig. 1K) and western blotting (Fig. 1L), which led to significantly reduced radiolabelled glutamine uptake (Fig. 1M). Similar to the single-cell clones, we observed no significant difference in growth between the HCC1806 non-targeted control (NC) and ASCT2 knockout (A2KO) lines, suggesting that the growth adaption in the A2KO lines were not due to CRISPR/Cas9 transfection or the single-cell cloning procedures (Fig. 1N). To further explore the lack of a growth phenotype, we also performed a colony formation assay and found no significant difference in the average size of colonies (Fig. 1O). We also assessed the effect of reducing glutamine concentration on cell growth of A2KO cells and found no significant difference in growth between HCC1806 NC and A2KO cells (Fig. EV1A). These data, however, clearly showed that the A2KO cells remained glutamine addicted and were unable to grow in the absence of glutamine (Fig. EV1A).

We next expanded the ASCT2 knockout analysis to additional TNBC cell lines, MDA-MB-231 and MDA-MB-468, which have been previously shown to be reliant on ASCT2 and glutamine (Gross et al, 2014; Timmerman et al, 2013; Van Geldermalsen et al, 2016). Similar to HCC1806 cells, we detected complete loss of ASCT2 protein expression and impaired glutamine uptake in both MDA-MB-231 (Fig. 1P,Q) and MDA-MB-468 A2KO cell lines (Fig. EV1B,C). While the growth rates of the MDA-MB-231 (Fig. 1R) and MDA-MB-468 (Fig. EV1D) A2KO cell lines were similar to the control line by CCK8 assay, the average colony size was significantly reduced in both MDA-MB-231 (Fig. 1S) and MDA-MB-468 (Fig. EV1E) A2KO cell lines. These data suggest that while MDA-MB-231 and MDA-MB-468 cell lines (like HCC1806) show no proliferative phenotype, they are slower to compensate for the loss of ASCT2 when seeded at low densities. To determine whether there were any changes in the Luminal A phenotype, we also generated ASCT2 knockout lines in two Luminal A breast cancer cell lines, MCF7 and T47D (Fig. EV1F–M). As expected from this glutamine-independent subtype, despite efficient knock-out of ASCT2 (Fig. EV1F,G,J,K), we found no difference in cell growth or colony formation when compared to the CRISPR/Cas9 NC control line (Fig. EV1H,I,L,M). Finally, to further validate the growth adaption observed when ASCT2 is absent, we used a CRISPR guide that targets a different exon (exon 4) of ASCT2 in HCC1806 cells (Fig. EV1N). This ASCT2 knockout (A2KO#2) showed a similar phenotype to our original ASCT2 knockout (A2KO) in terms of protein expression (Fig. EV1O), glutamine uptake (Fig. EV1P) and sustained cell growth comparable to the NC#2 cells (Fig. EV1Q). Taken together, these data indicate that loss of ASCT2, from single-cell clones or polyclonal populations, does not have an impact on cell viability even though glutamine

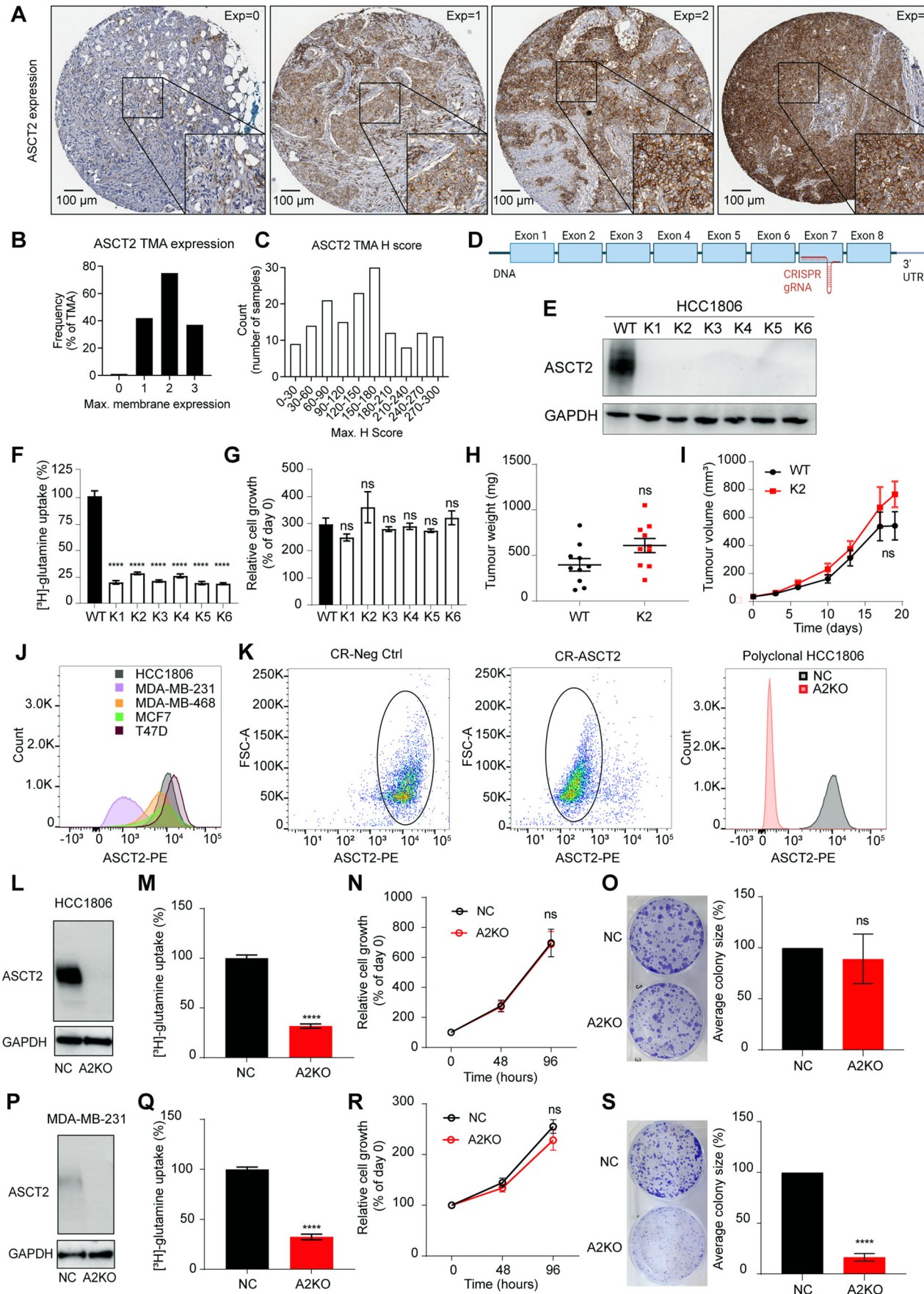

◄ **Figure 1. Growth adaption observed in TNBC cells to the loss of ASCT2.**

(A) Tissue microarray analysis (TMA) for ASCT2 (HPA035240, 1:2000, Sigma) protein expression in triple-negative breast cancer (TNBC) sections from 155 patients were scored based on the staining intensity from 0 to 3 as shown by the representative images. (B) ASCT2 expression scores for 154 samples were between 1 and 3, and only one sample scored 0, indicating no ASCT2 protein expression. (C) TMA H-score shows the spread of ASCT2 expression across samples. (D) Schematic for *SLC1A5* gene (ASCT2) showing 8 exons (boxes), introns (lines), as well as the CRISPR guide target (not to scale). (E) Western blot expression of ASCT2 (CST #8057S, 60–80 kDa) and GAPDH protein (Abcam #ab8245, 37 kDa) in HCC1806 wildtype (WT) and ASCT2 knockout single-cell clones (K1–6) showed complete loss of ASCT2 expression. (F) [³H]-L-glutamine uptake in HCC1806 WT and K1–K6 ASCT2 KO cell lines. Error bars are mean ± SEM from three independent experiments. Asterisks indicate *p* values, ****$p < 0.0001$ from a one-way ANOVA with Dunnett's multiple comparisons test. (G) Relative cell growth was measured by MTT assay in HCC1806 WT and K1–K6 ASCT2 KO cell lines after 3 days of cell growth and normalised to the respective day 0 MTT quantification for each cell line. Error bars are mean ± SEM from three independent experiments. No significant (ns) difference between WT and K1–K6 from a one-way ANOVA with Dunnett's multiple comparisons test. (H, I) HCC1806 orthotopic xenograft model tumour weights (endpoint) and tumour volume over time. Ten tumours per cohort (*n* = 5 mice/group, 2 tumours per mouse) with no significant (ns) difference from an unpaired *T*-test (J) Histogram of ASCT2 surface expression in breast cancer cell lines, analysed by flow cytometry. Cells were stained with primary antibody anti-ASCT2 antibody (MedImmune) and secondary antibody goat anti-human IgG Fc-PE secondary antibody (Invitrogen). (K) Sorting strategy for HCC1806 cells transfected with CRISPR CR-Neg Ctrl (non-targeted control; NC) or CR-ASCT2 based on ASCT2 surface expression and post-sort histogram of ASCT2 expression in expanded polyclonal HCC1806 NC and A2KO cells. (L, P) Polyclonal HCC1806 and MDA-MB-231 NC and A2KO cell lines were assessed by western blot for ASCT2 protein (CST #8057S, 60–80 kDa) with GAPDH protein (Abcam #ab8245, 37 kDa) as a loading control. (M, Q) Cell uptake of 100 nM [³H]-L-glutamine over 30 min. Error bars are mean ± SEM from three independent experiments. ****$p < 0.0001$ from an unpaired *T*-test. (N, R) Cells were seeded at a density of $2 \times 10^3$ (HCC1806) and $5 \times 10^3$ (MDA-MB-231) per well, with cell growth measured via CCK8 assay, normalised to day 0. Error bars are mean ± SEM from three independent experiments. No significant (ns) difference between NC and A2KO determined by two-way ANOVA. (O, S) Colony formation assay (CFA) in a six-well plate at $1 \times 10^3$ cells (HCC1806) and $2.5 \times 10^3$ (MDA-MB-231) per well, fixed and stained with 0.5% crystal violet after 12–14 days. Error bars are mean ± SEM from three independent experiments with ns (O) and ****$p < 0.0001$ (S) from unpaired *T*-tests. Source data are available online for this figure.

uptake into these cells seems to be dramatically reduced, and despite the cells remaining dependent on glutamine.

## Glutamine oxidation is altered in HCC1806 ASCT2 KO cells

To understand how metabolic activity is altered by the loss of ASCT2, we used the Seahorse XF assay to determine glutamine and glucose dependencies. First, we investigated glutamine dependence in the polyclonal A2KO cells by measuring the oxygen consumption rate (OCR) with and without glutamine (Fig. 2A). HCC1806 A2KO cells have a significantly increased baseline OCR compared to NC cells in the presence of glutamine, while both A2KO and NC cells exhibited a significant reduction in OCR in the absence of glutamine (Fig. 2A). This shows the reliance of HCC1806 cells on glutamine as a fuel, but also that there is an increased reliance on glutamine oxidation in the A2KO cells. This reliance was further confirmed using the Seahorse bioanalyser to inject glutamine (Fig. 2B), with a significant increase in OCR after the addition of glutamine, and again a significant increase in OCR in A2KO compared to NC (Fig. 2C). Interestingly, the addition of BPTES, an inhibitor of glutamine oxidation via glutaminase only slightly decreased the OCR (Fig. 2B). This suggests glutamine carbons may be entering the TCA cycle via conversion to glutamate/α-KG through additional pathways such as purine or pyrimidine biosynthesis enzymes (e.g. GMPS, CAD, ASNS, PPAT etc) which are all significantly increased in the TNBC subtype (Quek et al, 2022). After glutamine injection, ECAR reduced by a similar amount in both NC and A2KO cells, indicating a shift from glycolysis to glutaminolysis (Fig. 2D). Taken together, these data suggest that HCC1806 cells are overcompensating for the loss of ASCT2 by increasing glutaminolysis, thereby maintaining their reliance on glutamine.

To determine whether glucose dependence is altered with the loss of ASCT2, we also measured baseline OCR in the absence of glucose (Fig. 2E). Interestingly, OCR was significantly increased in HCC1806 NC and A2KO cells in the absence of glucose, suggesting these cells exhibit an increased capacity to efficiently utilise

glutamine at limiting glucose concentrations. However, once again, A2KO cells showed a significant increase in their capacity to utilise glutamine compared to NC cells (Fig. 2E). Injection of glucose significantly reduced OCR, but maintained the significant overall level of OCR in A2KO compared to NC cells (Fig. 2F,G). This reduction was due to a significant increase in ECAR, as the cells switch on glycolysis after glucose injection, however this was similar between both NC and A2KO cells (Fig. 2H). These data show the flexibility of the HCC1806 cells in utilising both glutamine and glucose, but clearly show the A2KO cells exhibit an increased glutamine carbon utilisation for TCA cycle compared to NC cells.

We also assessed the rate of glutamine oxidation in MDA-MB-231 cells, but found no difference in A2KO OCR at baseline or in media lacking glutamine (Fig. 2I). There was a significant decrease in OCR comparing baseline and glutamine-free media for both NC and A2KO cells (Fig. 2I), with glutamine injection (Fig. 2J) confirming that while MDA-MB-231 cells do not oxidise glutamine carbons to the same extent as HCC1806, glutamine is still an important fuel for OCR in MDA-MB-231 cells. Despite no substantial changes in OCR, ECAR was reduced in A2KO throughout the time course (Fig. 2K), with a significant reduction in ECAR in baseline A2KO cells compared to NC in MDA-MB-231 cells (Fig. 2L). Similarly, there were no significant changes in OCR or ECAR in the absence of glucose for A2KO, with only a significant increase in ECAR when glucose was injected in both NC and A2KO as expected (Fig. EV2A–D). These data suggest that while HCC1806 are clearly more glutamine dependent, both HCC1806 and MDA-MB-231 cells are still able to efficiently utilised glutamine carbons as an oxidative fuel, even in the absence of ASCT2.

## Intracellular glutamine and citrate levels are higher in A2KO cells

Since glutamine oxidation is still being efficiently utilised in A2KO cells, we used ¹³C₅-glutamine to trace the incorporation of these carbons into TCA intermediates and amino acids by performing

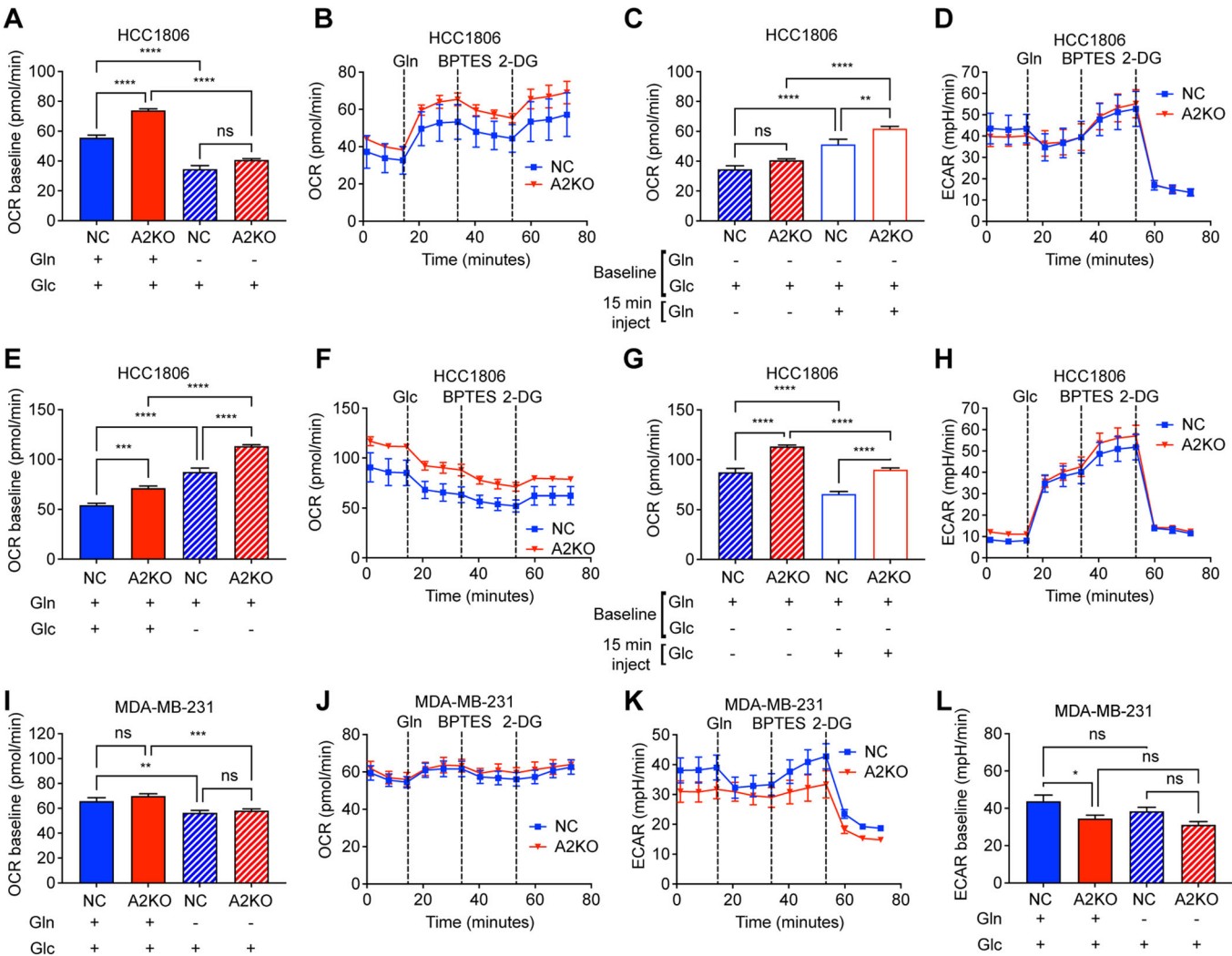

**Figure 2. Glutamine oxidation is altered in HCC1806 A2KO cells.**

Modified Seahorse XF assay was used to determine oxygen consumption rate (OCR) and extracellular acidification rate (ECAR) for HCC1806 (**A–H**) and MDA-MB-231 (**I–L**) polyclonal NC and A2KO cells. (**A**) OCR was measured in HCC1806 cells over 14 min in complete media or media lacking glutamine, ****$p < 0.0001$ (**B–D**) Seahorse assay was carried out in glutamine-free media, with sequential acute injection of 2 mM glutamine, followed by 3 µM BPTES (glutaminase inhibitor) and finally 50 mM 2-DG (glycolysis inhibitor). OCR (**B**) was measured, and OCR at baseline (first three readings) was compared to post glutamine injection (readings 5 and 6; **C**), $p$ values for (**C**) ****$p < 0.0001$, **$p = 0.0059$. ECAR (**D**) was used to measure glycolysis at each time point. (**E**) OCR was measured in HCC1806 cells over 14 min in complete media or media lacking glucose, ****$p < 0.0001$, ***$p = 0.002$. (**F–H**) Seahorse assay was carried out in glucose-free media, with a sequential acute injection of 11.1 mM glucose, followed by 3 µM BPTES (glutaminase inhibitor) and finally 50 mM 2-DG (glycolysis inhibitor). OCR (**F**) was measured, and OCR at baseline (first three readings) was compared to post glucose injection (readings 5 and 6; **G**), $p$ values for (**G**) ****$p < 0.0001$. ECAR (**H**) was used to measure glycolysis at each time point. (**I**) OCR was measured in MDA-MB-231 cells over 14 min in complete media or media lacking glutamine, ***$p = 0.0007$, **$p = 0.0079$. (**J–L**) Seahorse assay was carried out in glutamine-free media, with a sequential acute injection of 2 mM glutamine, followed by 3 µM BPTES (glutaminase inhibitor) and finally 50 mM 2-DG (glycolysis inhibitor). OCR (**J**) and ECAR (**K**) were measured at each time point. ECAR (**L**) was measured in MDA-MB-231 cells over 14 min in complete media or media lacking glutamine, $p$ values for (**L**) *$p = 0.0343$. Data are mean ± SEM from three independent experiments with 4–6 repeats per treatment, analysed by one-way ANOVA with Šídák's multiple comparisons test (ns not significant). Source data are available online for this figure.

metabolite extraction 24 h (steady state) after adding the tracer (Figs. 3A and EV3A). Interestingly, despite a significant reduction in the rate of glutamine uptake (Fig. 1F), intracellular $^{13}C_5$-labelled glutamine levels were significantly higher in both HCC1806 and MDA-MB-231 A2KO cells (Fig. 3A). In HCC1806 A2KO cells, both $^{13}C$-labelled isotopologues (m5 and m3) and the unlabelled glutamine isotopologue (m0) were significantly increased, indicating that in these cells, glutamine supplies are being replenished by another source in addition to extracellular $^{13}C$-labelled glutamine such as de novo

glutamine synthesis or by blocking the efflux of intracellular glutamine. Glutamate and α-ketoglutarate (α-KG) levels varied between the A2KO cell lines as compared to the respective NC cell lines, with the majority being derived from one or more $^{13}C$-glutamine carbons. HCC1806 A2KO cells showed higher unlabelled and total glutamate levels, while in MDA-MB-231 A2KO cells, glutamate levels were similar to the NC cells (Fig. 3B). α-KG levels showed an inverse relationship to glutamate levels, i.e. in HCC1806 A2KO cells α-KG (m5) was lower than in NC cells and in MDA-MB-231 A2KO cells

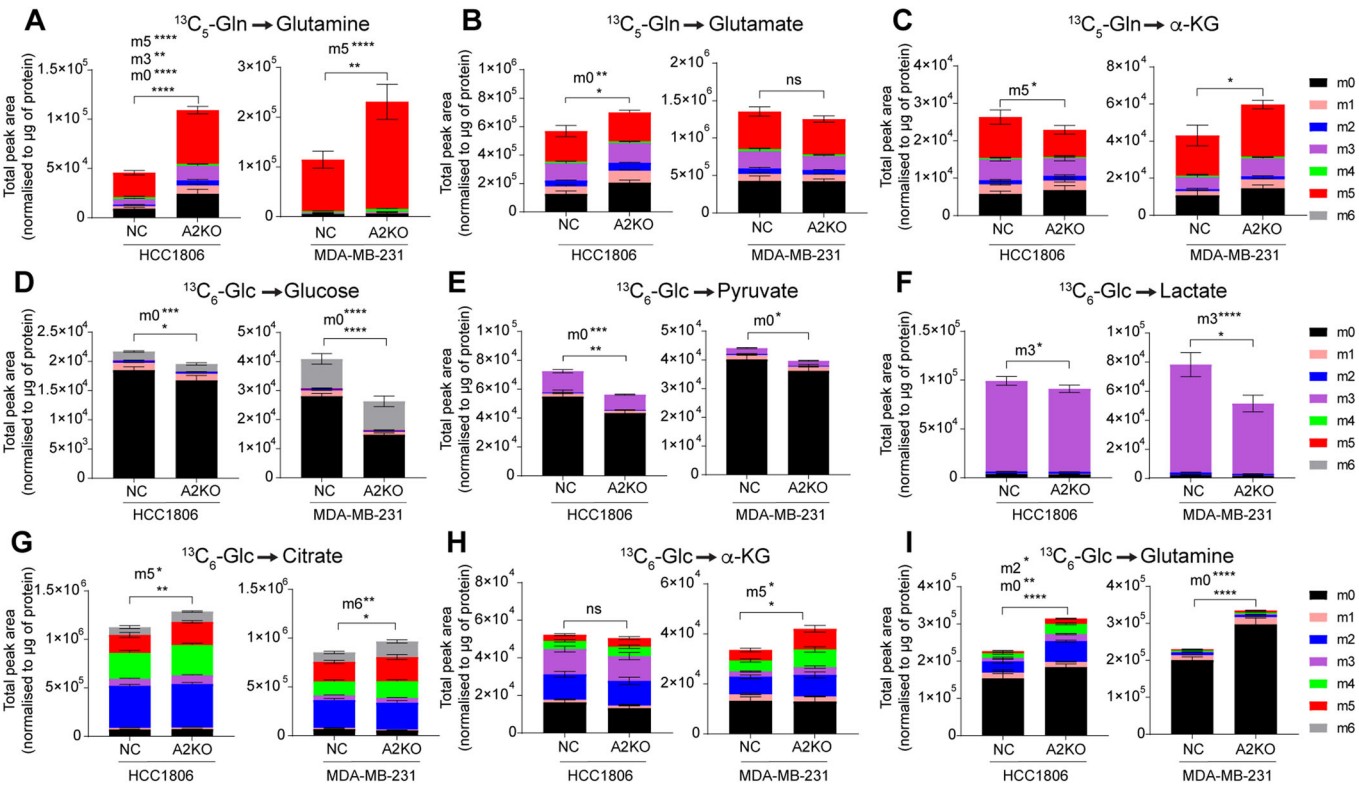

**Figure 3. Glutamine uptake and de novo glutamine synthesis were higher in HCC1806 A2KO cells.**

(A–C) Metabolomics was used to trace $^{13}C_5$-glutamine ($^{13}C_5$-Gln) carbons in polyclonal HCC1806 and MDA-MB-231 NC and A2KO cells. The total peak area of intracellular glutamine, glutamate and α-KG derived from $^{13}C_5$-glutamine were detected by LC-MS at 24 h. *p* values for differences between NC and A2KO cells for (A) glutamine in HCC1806 ****$p < 0.0001$, m0 ****$p < 0.0001$, m3 **$p = 0.009$, m5 ****$p < 0.0001$ and MDA-MB-231 ** $p = 0.0044$, m5 ****$p < 0.0001$, (B) glutamate in HCC1806 *$p = 0.0197$, m0 **$p = 0.0041$ and MDA-MB-231 not significant (ns) for all, (C) α-KG in HCC1806 m5 *$p = 0.0153$ and MDA-MB-231 *$p = 0.0226$. (D–I) Metabolomics was used to trace $^{13}C_6$-glucose ($^{13}C_6$-Glc) carbons in polyclonal HCC1806 and MDA-MB-231 NC and A2KO cells. *p* values for differences between NC and A2KO cells for (D) glucose in HCC1806 *$p = 0.0440$, m0 ***$p = 0.0002$ and MDA-MB-231 ****$p < 0.0001$, m0 ****$p < 0.0001$, (E) pyruvate in HCC1806 **$p = 0.0025$, m0 ***$p = 0.0002$ and MDA-MB-231 m0 *$p = 0.0477$, (F) lactate in HCC1806 m3 *$p = 0.0428$ and MDA-MB-231 *$p = 0.0114$, m3 ****$p < 0.0001$, (G) citrate in HCC1806 **$p = 0.0018$, m5 *$p = 0.0489$ and MDA-MB-231 *$p = 0.0265$, m6 **$p = 0.0088$, (H) α-KG in HCC1806 ns for all and MDA-MB-231 *$p = 0.0452$, m5 *$p = 0.0224$, (I) glutamine in HCC1806 ****$p < 0.0001$, m0 **$p = 0.0062$, m2 *$p = 0.0145$ and MDA-MB-231 ****$p < 0.0001$ and m0 ****$p < 0.0001$. The total peak area of intracellular glucose, pyruvate, lactate, citrate, α-KG and glutamine were detected by LC-MS at 24 h. Mass of the unlabelled metabolite ($^{12}C = m0$), which changes with integration with $^{13}C$-labelled carbons, where (m#) indicates a metabolite with # of carbons labelled with $^{13}C$. (m0) isotopologue denotes that all carbons of the metabolite are $^{12}C$, and the metabolite is unlabelled. (m5) isotopologue signifies that five carbons are $^{13}C$ isotopes ($^{13}C_5$-Gln). (m6) isotopologue signifies that 6 carbons are $^{13}C$ isotopes ($^{13}C_6$-Glc). The total peak area was normalised to μg of protein for each sample. Error bars are mean ± SEM from three independent experiments performed in duplicate, analysed by two-way ANOVA. Where isotopologues are listed with asterisks, the unlisted isotopologues are ns. Source data are available online for this figure.

total α-KG levels were higher (Fig. 3C), suggesting these cells differentially balance glutamine/glutamate/α-KG interconversion, before entering into TCA cycle.

## HCC1806 A2KO cells utilise glucose carbons for glutamine synthesis

We next used $^{13}C_6$-glucose to trace the incorporation of glucose carbons into glycolysis and TCA intermediates by performing metabolite extraction 24 h (steady state) after adding the tracer (Fig. EV3A). Using metabolomics analyses, we observed that glucose, pyruvate, and lactate levels were significantly lower in A2KO cells (Fig. 3D–F). Pyruvate gets converted to acetyl-CoA, which enters the TCA cycle and contributes two carbons to citrate. Total and labelled citrate levels were significantly higher in HCC1806 and MDA-MB-231 A2KO cells (Fig. 3G). Glucose

carbons contributed to a substantial proportion of α-KG in both cell lines, with a significant increase seen in MDA-MB-231 cells (Fig. 3H). By comparison, there were no substantial alterations in other TCA metabolites such as succinate, fumarate, or malate levels in either cell line, which again consisted of mainly glucose-derived carbons (Fig. EV3B–D).

De novo glutamine synthesis via glucose catabolism can utilise TCA-derived α-KG to generate glutamate, which can then be combined with ammonia to generate glutamine (Cruzat et al, 2018). Both cell lines showed a high abundance of both glucose and glutamine $^{13}C$-labelled glutamate and α-KG levels, suggesting the mixing of TCA-derived carbons in and out of the TCA cycle at α-KG (Figs. 3B,C,H and EV3E). Furthermore, in HCC1806 A2KO cells, unlabelled and labelled glutamine levels were higher than in NC cells, while MDA-MB-231 A2KO cells consisted predominantly of unlabelled glutamine, which was higher than its matched control

(Fig. 3A,I). These data suggest that in HCC1806 A2KO cells, high glutamine levels are maintained by increasing glutamine uptake and an increase in the synthesis of glutamine by catabolising glucose via α-KG in the TCA cycle. On the other hand, in MDA-MB-231 A2KO cells, glutamine levels only seem to be enhanced by increasing glutamine uptake but not through glucose catabolism.

## Proteins involved in macropinosome formation are upregulated in HCC1806 A2KO cells

Based on the data above it is evident that HCC1806 A2KO cells are employing multiple mechanisms to compensate for the loss of ASCT2 and restore glutamine reserves. To better understand the mechanisms involved, we carried out proteomics (Dataset EV1) and mRNA sequencing (Dataset EV2) for HCC1806 WT and A2KO single-cell clones (K2 and K3). We included an additional A2KO single-cell clone (KO), from a previously published study for these analyses (Bröer et al, 2019). For proteomics analysis, we used HCC1806 WT and three single-cell clones (KO, K2 and K3) and identified top 50 downregulated and top 50 upregulated proteins in the ASCT2 knockouts (Fig. 4A, left and right). To annotate the pathways upregulated in A2KO cells, we used functional enrichment analysis which showed enrichment of several proteins that are associated with cellular compartments such as macropinosome (two out of nine proteins), ruffle (5 out of 209 proteins) and anchoring junction (15 out of 987 proteins; Fig. 4B). Next, we looked at proteins involved in macropinocytosis (GO_0044351) curated from gene set enrichment analysis (GSEA). Positive regulators of macropinocytosis, such as SRC, SNX9 and SNX12, as well as RAC1, required for the formation of macropinosome (West et al, 2000), were significantly upregulated in the A2KO clones (Fig. 4C). We confirmed the increased protein expression of SRC and SNX9 by western blot (Fig. 4D). APPL1, an inhibitor of SRC activity (Broussard et al, 2012), and PAK4, an actin cytoskeleton re-modeller (Won et al, 2019), were significantly downregulated in A2KO clones. Gene expression analyses of mRNA sequencing data revealed SRC was significantly upregulated while *PAK4* was downregulated in A2KO cells, consistent with proteomics data (Fig. EV4A). Interestingly, the mRNA expression of *ROBO1*, a negative regulator of macropinocytosis (Bhosle et al, 2020), was downregulated albeit not significantly, but ROBO1 peptides were absent in the proteomics data.

Next, we examined whether the complete loss of ASCT2 alters the protein and gene expression of amino acid transporters. As expected, ASCT2 (SLC1A5) protein was significantly downregulated across all three A2KO clones (Fig. 4E). The glutamine transporter SLC38A1 (SNAT1) was upregulated in the previously published A2KO clone (but not in K2 or K3 clones), which may play a role in increasing glutamine levels (Fig. 4E). SLC7A5 (LAT1), a leucine/glutamine exchanger, was significantly downregulated in all 3 A2KO clones (Fig. 4E) suggesting that another compensation mechanism may be reducing glutamine export, thereby increasing intracellular glutamine levels (Bhutia and Ganapathy, 2016). In addition, SLC7A11 (xCT), cystine/glutamate antiporter, was upregulated in A2KO clones (Fig. 4E). This transporter plays an important role in defence against oxidative stress as it imports cystine (in exchange for glutamate) which is needed for glutathione biosynthesis. SLC7A11 also inhibits ferroptosis, an iron-dependent programmed cell death pathway, thereby aiding in cell survival (Koppula et al, 2021).

We also analysed the protein expression of metabolic enzymes required for the TCA cycle. Enzymes involved in the synthesis of succinate (succinyl-CoA synthase, SUCLA2 and SUCLG1), fumarate (succinate dehydrogenase, SDHB), malate (fumarase, FH) and citrate (citrate synthase, CS) were significantly downregulated in A2KO clones (Fig. 4F). In addition, oxoglutarate dehydrogenase (OGDH) and dihydrolipoyl dehydrogenase (DLD), two components of the enzyme complex that converts α-KG to succinyl-CoA were also consistently downregulated across the A2KO clones, which may assist with building up α-KG and subsequent exit from the TCA cycle. There was also differential expression of enzymes that convert glutamine to glutamate in de novo purine synthesis, including downregulation of GMP synthase (GMPS) and upregulation of phosphoribosyl pyrophosphate amidotransferase (PPAT; Fig. 4F). In addition, a number of glycolytic enzymes were consistently and significantly upregulated in A2KO clones, including enzymes such PGK1, Enolase and PKM which would result in increased utilisation of glucose carbons to generate pyruvate (Fig. 4F).

On analysing the gene expression profile of amino acid transporters in A2KO lines, we found no significant increase in members of the *SLC38* family (Bhutia and Ganapathy, 2016) (Fig. EV4B). In addition, we also compared the expression of *SLC6A14* from the neurotransmitter family that is often upregulated and can transport glutamine in some cancer subtypes (Karunakaran et al, 2011; Bröer et al, 2011) and found no increase in transcript expression in the knockout lines (Fig. EV4B). While *SLC7A5* gene expression in the knockout lines was unaffected by the loss of ASCT2, gene expression of another leucine/glutamine antiporter *SLC7A8* was reduced across the knockout lines (Fig. EV4B) though SLC7A8 (LAT2) protein was not detected in our proteomics analysis. Additionally, we observed no change in the expression of glucose transporters *SLC2A1*, *SLC2A2*, *SLC2A3* or *SLC2A4* (Fig. EV4B). These results indicate that HCC1806 cells compensate for the loss of ASCT2 by altering multiple pro-survival mechanisms such as upregulation of macropinocytosis, inhibition of ferroptosis and reduction of glutamine efflux. Additionally, these alterations seem to be predominantly regulated at the protein level rather than the transcript level.

Since the A2KO cells showed an opposing growth phenotype to the previously published study on the HCC1806 ASCT2 knockdown (shASCT2) cells, we also included these same shCtrl and shASCT2 cells in our gene expression analysis (Van Geldermalsen et al, 2016). We first confirmed the phenotype of these cells, with the shASCT2 knockdown reducing ASCT2 expression (Fig. EV4C), inhibiting glutamine uptake (Fig. EV4D) and showing significant growth inhibition by both CCK8 assay (Fig. EV4E) and Incucyte analyses (Fig. EV4F), confirming previous data (Van Geldermalsen et al, 2016). We used Metascape to identify pathways enriched in HCC1806 shASCT2 (KD) and KO cells and found a differential set of pathways were enriched between the KD and KO lines, supporting the divergence in growth phenotype observed when compared to the KO cells (Fig. EV4G).

## Macropinocytosis is upregulated in HCC1806 A2KO cells

Due to the upregulation of proteins involved in membrane ruffling and macropinosome formation in HCC1806 A2KO cells, we examined if this resulted in an increase in macropinocytosis using

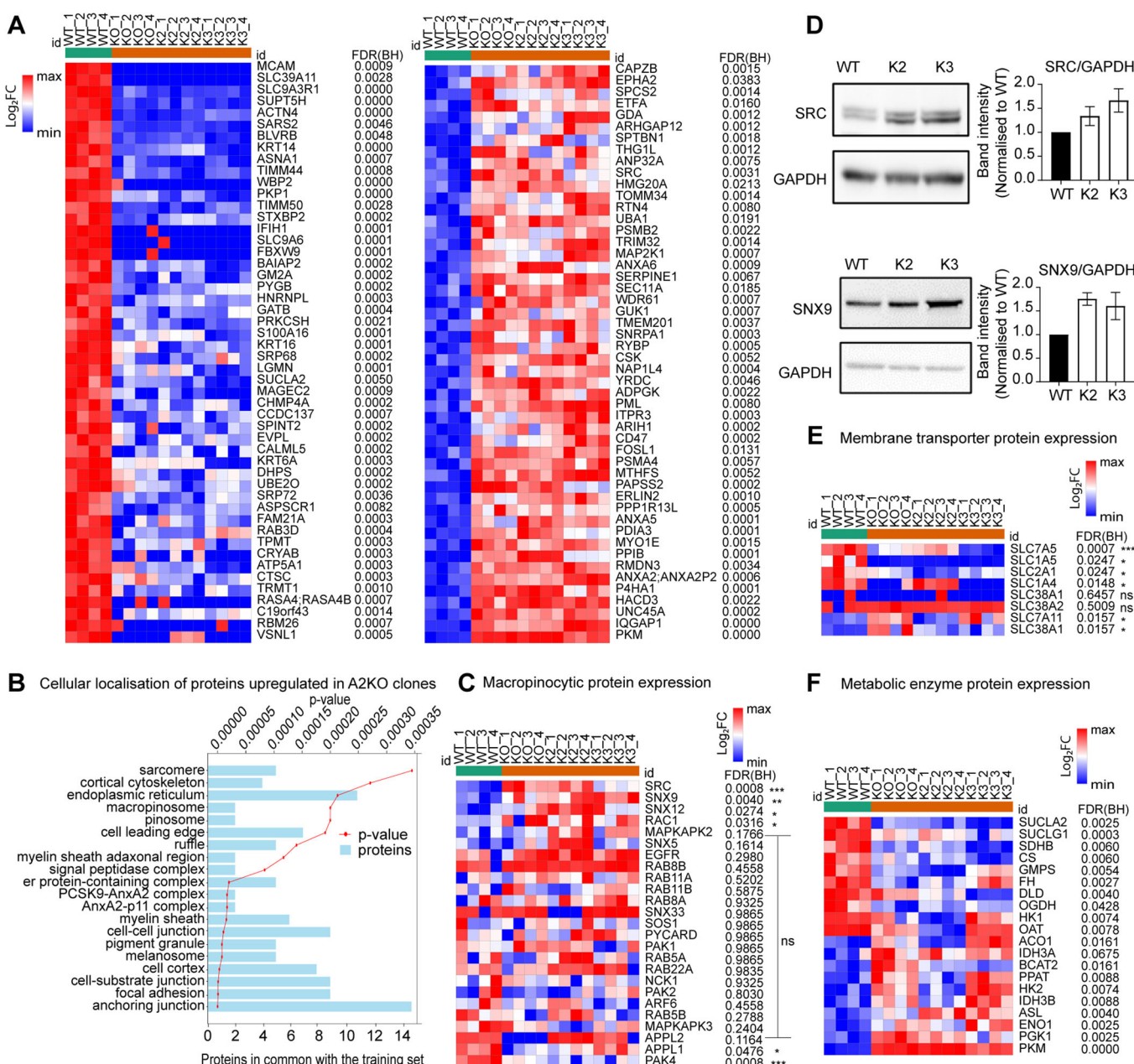

**Figure 4. Macropinocytic and glycolytic proteins are upregulated in HCC1806 A2KO cells.**

(A) Heatmaps for the top 50 differentially upregulated and downregulated proteins in three single-cell ASCT2 knockout clones (KO, K2 and K3) compared to HCC1806 WT, generated using Morpheus online software (Broad). Protein expression between WT and the 3 KO clones (n = 4 biological replicates) was analysed by unpaired t-test, and all listed proteins were statistically significant (FDR, false discovery rate). (B) Cellular localisation of the top 50 upregulated proteins in the KOs was determined using Toppfun online portal where proteins in common with the annotated proteins in each cellular compartment are shown as blue bars and the p value (red) represents the probability that the upregulated proteins shared with the annotated list are occurring by chance. The first 20 cellular compartments with the lowest p values are shown as determined by "Bonferroni" for the multiple correction method at 0.05 cut-off level for significance. (C) Heatmap for expression of proteins involved in macropinocytosis where asterisks indicate p value from unpaired t-test (*p ≤ 0.05, **p ≤ 0.01, ***p ≤ 0.001, ns is not significant). (D) Western blots for SRC (CST #2108S, ~60 kDa) and SNX9 (Abcam #ab181856, ~70 kDa) with GAPDH loading control (Abcam #ab8245, 37 kDa) and quantification of band intensity. Data are mean ± SEM band intensities from five independently collected lysates. (E, F) Heatmaps for expression of proteins involved in amino acid and glucose transport where all listed proteins are statistically significant (E) and metabolic enzymes involved in glutamine and glucose metabolism (F) where asterisks indicate p value from unpaired t-test (*p ≤ 0.05, ***p ≤ 0.001, ns is not significant). A relative colour scheme is used for heatmaps (A, C, E, F) where minimum (blue) and maximum (red) values in each row are converted to colours (https://software.broadinstitute.org/morpheus/). Source data are available online for this figure.

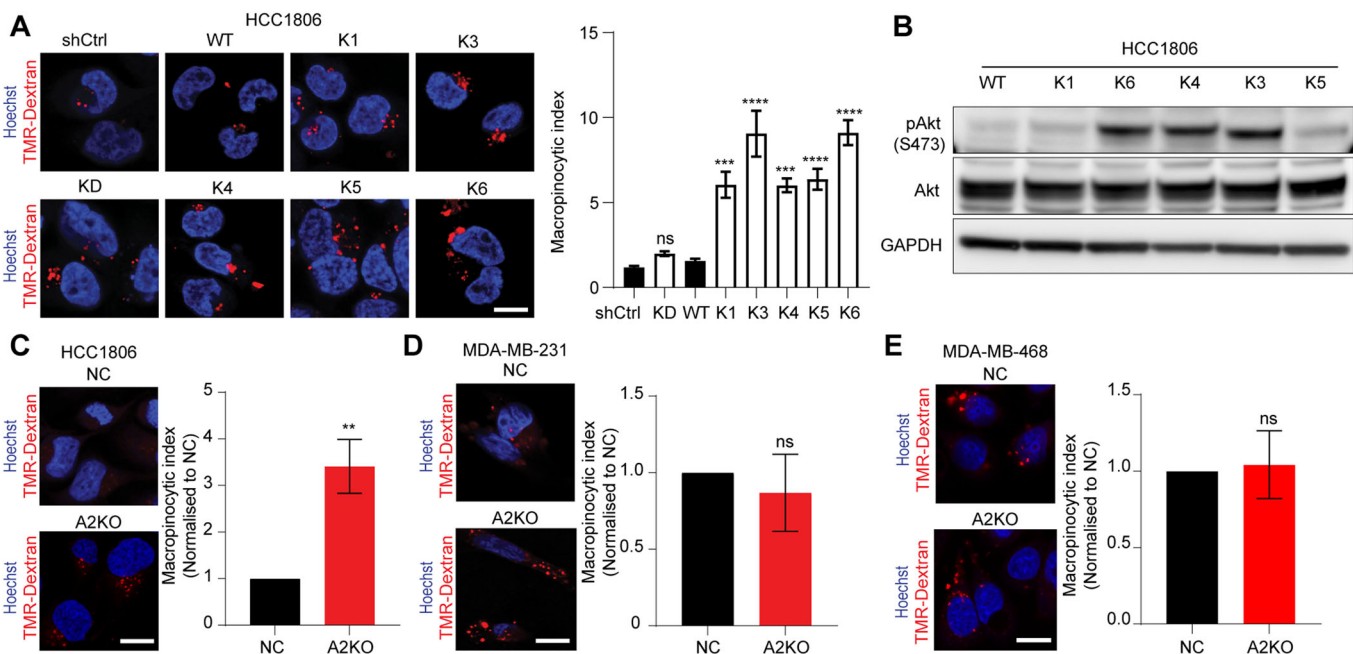

**Figure 5. Macropinocytosis is induced by the loss of ASCT2 in HCC1806 but not in MDA-MB-231.**

(A) Uptake of TMR-dextran (red) in HCC1806 shCtrl and ASCT2 KD, and WT and ASCT2 KO single-cell clones. Nuclei were visualised with Hoescht fluorescent stain (blue). The scale bar in white (K6 panel) is equivalent to 10 μM. shCtrl and KD were pre-incubated for 72 h in media + 1 μg/mL dox (as indicated) prior to uptake. Quantification of macropinocytic index in HCC1806 cell lines is macropinocytosis/cell (area of macropinosomes/number of cells in the field; SEM from 9–12 fields across three separate experiments). Asterisks indicate *p* value from a one-way ANOVA with Dunnett's multiple comparisons test with WT and K1 ***p = 0.0003, WT and K4 ***p = 0.0004, WT and K3, K5, K6 ****p < 0.0001. (B) Western blot of pAKT (S473) protein (CST #9271, 60 kDa), Akt (CST #9272, 60 kDa) and GAPDH protein (Abcam #ab8245, 37 kDa) as a loading control in HCC1806 WT and single-cell ASCT2 KO clones. Representative images from two independent repeats are shown. (C–E) Uptake of TMR-dextran (red) in polyclonal HCC1806, MDA-MB-231 and MDA-MB-468 NC and A2KO cell lines. Nuclei were visualised with Hoescht fluorescent stain (blue). The scale bar in white is equivalent to 10 μM. Quantification of macropinocytic index in A2KO cell lines is macropinocytosis/cell (area of macropinosomes/number of cells in the field) and normalised to NC cells. Data are mean ± SEM of indices calculated from five fields of view from three (D, E) or four (C) independent experiments. Asterisks indicate *p* value from an unpaired *T*-test comparing NC and A2KO cells in (C) HCC1806 **p = 0.0059 and (D, E) MDA-MB-231 and MDA-MB-468 ns (not significant). Source data are available online for this figure.

fluorescently labelled 70 kDa dextran (Commisso et al, 2014). While HCC1806 cells exhibit macropinocytosis in WT and shCtrl cells, there was a significant increase in the macropinocytic index across the HCC1806 single-cell knockout clones (Fig. 5A). By comparison, the knockdown cells did not show a significant increase in macropinocytosis (Fig. 5A).

Akt phosphorylation at serine 473 (S473) (Michalopoulou et al, 2020) and AMPK phosphorylation (Kim et al, 2018) regulate membrane ruffling and macropinosome formation by activating proteins such as Rac1 (Araki et al, 1996; Amyere et al, 2000; Kim et al, 2018). We found that Akt S473 phosphorylation was substantially higher in the majority of the knockout clones, which may contribute to the increased macropinocytosis (Fig. 5B). We also observed an increase in macropinocytosis in the polyclonal HCC1806 knockout cells which also exhibited increased AMPK and Akt phosphorylation compared to the NC cells (Figs. 5C and EV5A). These data indicate that loss of ASCT2 in HCC1806 cells is associated with an upregulation of macropinocytosis, which could be the alternate route by which these cells acquire nutrients such as glutamine. The lack of macropinocytosis upregulation in the ASCT2 knockdown cells may explain why they are unable to compensate for ASCT2 loss, resulting in significant growth inhibition compared to control (Fig. EV4E,F).

Macropinocytosis is a non-specific mechanism mediating the uptake of extracellular components, and we next sought to clarify the mechanism by which macropinocytosis rescues the A2KO cells, and whether this process relied on free amino acid uptake, or on protein breakdown. We first attempted to rescue cell growth using BSA as a protein source in glutamine-free conditions, since BSA could be broken down to produce amino acids after fusion of the macropinosome to lysosomes. Similar to previously published data (Jayashankar and Edinger 2020), free protein (BSA) was unable to rescue HCC1806 KO cell growth in the absence of free glutamine (Fig. EV5B). Using de-quenched BSA (DQ-BSA) and lysotracker co-localisation analysis, we confirmed that macropinosome/lysosome fusion was occurring, however there were minimal changes in co-localisation between WT and KO cells (Fig. EV5C). Similarly, there was no difference in WT or KO cells treated with the lysosomal inhibitor chloroquine (Fig. EV5D), further confirming that protein breakdown was not a substantial component of the KO cell resistance phenotype. By comparison, our ¹³C-glutamine tracing data (Fig. 3A) clearly suggest that extracellular free glutamine and, to some extent, free glucose, are the critical macropinocytosed nutrients being utilised by HCC1806 A2KO cells.

We next determined whether ASCT2 re-expression could revert the macropinocytic phenotype in A2KO cells. Overexpression of ASCT2 restored protein expression (Fig. EV5E) and glutamine

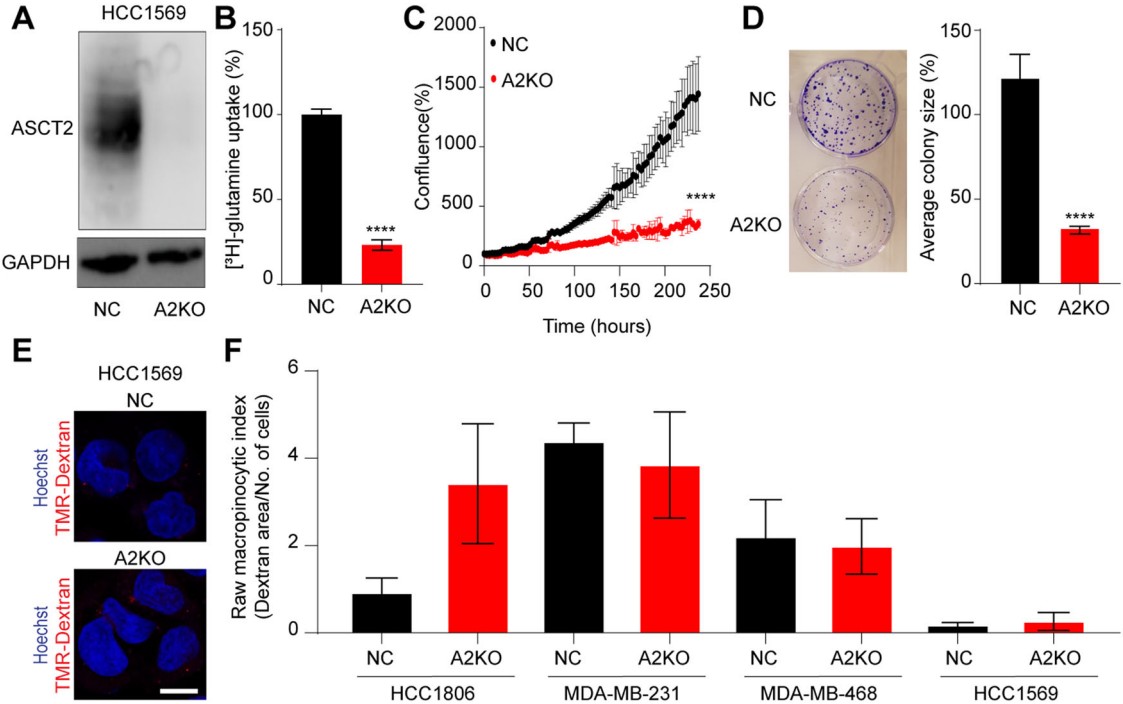

**Figure 6. ASCT2 impairs growth in the non-macropinocytic breast cancer cell line, HCC1569.**

(A) Western blot of ASCT2 protein (CST #8057S, 60–80 kDa) and GAPDH protein (Abcam #ab8245, 37 kDa) as a loading control in polyclonal HCC1569 (Her2+ cell line) NC and A2KO cell lines. (B) Uptake of 100 nM [³H]-glutamine in HCC1569 NC and A2KO cell lines over 30 min. Error bars are mean ± SEM from three independent experiments. ****$p < 0.0001$ from an unpaired $T$-test. (C) Growth of HCC1569 cells seeded at a density of $1 \times 10^4$ cells per well in a 12-well plate, measured by live cell imaging (IncuCyte® S3) over 240 h. Error bars are mean ± SEM from three independent experiments, each containing a single well due to limited cell numbers. ****$p < 0.0001$ from an unpaired $T$-test. (D) Colony formation assay (CFA) for HCC1569 cells in a six-well plate, fixed and stained with 0.5% crystal violet after 12–14 days. Error bars are mean ± SEM from three independent experiments. ****$p < 0.0001$ from an unpaired $T$-test. (E) Representative images of uptake of TMR-dextran (red) in HCC1569 NC and A2KO cells, and nuclei were visualised with Hoescht fluorescent stain (blue) from >5 fields of view across three independent experiments. The scale bar in white is equivalent to 10 μM. (F) Raw macropinocytic index across breast cancer cell lines. Error bars are mean ± SEM from three independent experiments. Data in (F) are mean ± SEM of indices calculated from >5 fields of view across three independent experiments. Source data are available online for this figure.

uptake (Fig. EV5F) in A2KO cells, and while this did not change proliferative capacity (Fig. EV5G), it reduced macropinocytosis levels back to control (Fig. EV5H), suggesting that intracellular glutamine levels drive this adaptive yet reversible process. Interestingly, we found no difference in macropinocytosis between the control cells and MDA-MB-231 or MDA-MB-468 A2KO cells (Fig. 5D,E). The phosphorylation of AMPK was reduced in MDA-MB-231 A2KO cells with no change in phospho-Akt as it is known to be at very low levels in the MDA-MB-231 cell line (Yang et al, 2011) (Fig. EV5I). In MDA-MD-468 A2KO cells, phosphorylation of both AMPK and Akt was reduced (Fig. EV5J). These two cell lines have previously been shown to exhibit a high macropinocytotic index compared to HCC1806 cells (Jayashankar and Edinger 2020), which may be sufficient to facilitate their continued growth after ASCT2 knockout.

By comparison, HCC1569, a Her2-positive breast cancer cell line, is known to be non-macropinocytic even under nutrient stress (Jayashankar and Edinger 2020). We therefore used HCC1569 cells to further elucidate how loss of ASCT2 effects cell survival. Using the CRISPR/Cas9 negative control and exon 7 guide RNA, we generated polyclonal HCC1569 NC and A2KO cell lines. ASCT2 protein knockout was confirmed by western blotting (Fig. 6A) and resulted in a significant reduction in glutamine uptake after 30 min

(Fig. 6B). Unlike the macropinocytic TNBC cell lines, there was a substantial reduction in cell growth assessed using Incucyte in A2KO HCC1569 (Fig. 6C), as well as a significant reduction in colony size (Fig. 6D). To assess macropinocytosis, we performed dextran uptake assays for HCC1569 NC and A2KO cells, confirming the previously published low basal macropinocytosis in NC cells, and showed no induction of macropinocytosis after ASCT2 knockout (Fig. 6E). Analysis of the macropinocytic index for these four cell lines suggests there is a threshold of macropinocytic activity that is required to facilitate resistance to A2KO-mediated loss of glutamine uptake, which is not reached for HCC1569 cell lines (Fig. 6F).

## Discussion

The glutamine transporter ASCT2 has been explored as a potential therapeutic target in a range of cancers (Wahi and Holst, 2019). ASCT2 is highly expressed in many cancers and cancer subtypes, with more restricted expression in normal tissues (Wang et al, 2015; Ansari et al, 2020; Nakaya et al, 2014; Ren et al, 2015; Hassanein et al, 2013). In addition, the utility of glutamine as both a carbon and nitrogen source for cancer cells makes it a critical

nutrient, which becomes conditionally essential in some cancers or cancer subtypes compared to normal cells (Wei et al, 2021a). However, its potential utility as a therapeutic target has been hampered by a lack of specific inhibitors of its activity. In addition, it is clear that there are multiple mechanisms present in cancer cells that may provide avenues for resistance to ASCT2 inhibition, such as those seen for glutaminase inhibitors (Dos Reis et al, 2019; Singleton et al, 2020), as well as the potential for upregulation of alternative glutamine transporters (Bröer et al, 2019). Recent reviews of this area have also highlighted the difficulties of targeting glutamine metabolism, including the potential for macropinocytosis as a resistance mechanism (Jin et al, 2023). Based on previous siRNA/shRNA data suggesting ASCT2 may be a good target in TNBC (Jeon et al, 2015; Van Geldermalsen et al, 2016), we set out to further explore how ASCT2 loss affects the growth and metabolism of TNBC cells.

TNBC patient tumours exhibited strong membrane expression of ASCT2 (Fig. 1A–C) on 154/155 TNBC samples, in accordance with previous *SLC1A5* (ASCT2) mRNA expression data on 96 patients in this same cohort (Van Geldermalsen et al, 2016). To address the biological effect of ASCT2 expression, we took a comprehensive approach to determine if ASCT2 is required for cell survival under different glutamine concentrations and across different glutamine-dependent (TNBC) and independent (Luminal A) breast cancer cell lines. While the loss of ASCT2 significantly inhibited glutamine uptake across all breast cancer cell lines, it did not affect cell growth either in vitro or in vivo models (Fig. 1). It was, however, intriguing to observe differences in colony-forming ability with the loss of ASCT2 across TNBC cell lines. HCC1806, derived from the primary site, showed no difference in colony-forming ability with the loss of ASCT2 while MDA-MB-231 and MDA-MB-468 cell lines, derived from a pleural effusion site, showed significantly smaller colonies in the knockout cells. This suggests there may be differences in metabolite requirements between cells from the primary tumour as opposed to a metastatic site. We and others have previously shown that Luminal A cell lines, despite having high ASCT2 expression and ASCT2-mediated glutamine uptake, are not glutamine addicted (Van Geldermalsen et al, 2016), and, as expected, showed no difference in colony-forming ability (Fig. EV1F–M).

One explanation for the lack of growth inhibition could be that the mutation caused by exon 7 CRISPR guide results in a partially functional protein or isoform that is not detectable by ASCT2 antibodies. We mapped the frameshift mutations (mRNAseq and Sanger sequencing) and used our proteomics data to show there were no ASCT2 peptides in KO cells upstream of the guide RNA (Appendix Fig. S1A–C). In addition, since targeting exon 4 also resulted in the same phenotype, this suggests that it is unlikely that a partially functional ASCT2 may be responsible for the growth phenotype observed in A2KO cells (Fig. EV1N–Q). Previous data in a pancreatic cancer model showed the existence of both the plasma membrane and a mitochondrial variant of ASCT2, which was suggested to be critical for glutamine-dependent mTORC1 activation required for cell survival (Yoo et al, 2020). Both ASCT2 isoforms have the same coding regions spanning from exon 3 to 8 and only differ at the N-terminus. Since the CRISPR guides we used target exon 4 or 7, they would knock out the expression of both the plasma membrane and mitochondrial isoforms in the A2KO cells, ruling out a role for the mitochondrial isoform in resistance. These

data suggest that adaptation at the cellular and/or metabolic level is responsible for maintaining growth in ASCT2 knockout cells.

The divergence in colony formation upon loss of ASCT2 between HCC1806 and MDA-MB-231 cells was observed in their metabolic activity as well. For sustained growth a balance between mitochondrial respiration and glycolysis is needed for energy requirements as well as to maintain TCA intermediates that generate amino acids for protein production. While the loss of ASCT2 in HCC1806 cells appears to boost glutamine oxidation, this phenomenon was not observed in MDA-MB-231 A2KO cells (Fig. 2). Additionally, the lower ECAR levels and the lack of drop in glycolysis in MDA-MB-231 A2KO cells when glutamine was injected in the Seahorse assay suggests that at least in the short term, the balance between mitochondrial respiration and glycolysis is impaired in these cells.

Our data also suggests that the loss of ASCT2 in HCC1806, but not MDA-MB-231, activates de novo glutamine synthesis from glucose, with $^{13}C$-glucose carbons being incorporated into glutamine in HCC1806 A2KO cells (Fig. 3). Glucose driven anaplerosis has been observed in breast cancer cells though predominantly in the Luminal A subtype (Quek et al, 2022). These previous data also showed that glutamine oxidation via single-pass glutaminolysis can drive glucose into the TCA cycle in HCC1806 cells, which also supports increased production of glutamate from α-KG (Quek et al, 2022). While our proteomics data did not show an increase in glutamine synthetase (GS/*GLUL*), the enzyme that converts glutamate to glutamine, it was expressed consistently across HCC1806 WT and A2KO lines and likely contributed to the glutamine production from glucose-derived glutamate (Dataset EV1).

Higher expression of glycolytic proteins (Fig. 4F), such as PGK1, ENO1 and PKM in HCC1806 A2KO cells suggests an increase in demand for pyruvate. PKM is the rate-limiting enzyme that determines the fate of the pyruvate being converted to lactate (Warburg, 1956) or if it is shunted into the TCA cycle as Acetyl-CoA (Zahra et al, 2020). Glucose tracing experiments showed higher levels of labelled citrate in HCC1806 A2KO cells (Fig. 3G), further suggesting that upon the loss of ASCT2, a larger proportion of glucose carbons are diverted into the TCA cycle. This, together with the reduction in enzymes pushing α-KG into TCA (Fig. 4F), could also contribute (via the α-KG to glutamate pathway) to the higher levels of $^{13}C$-labelled glutamine in HCC1806 A2KO (Fig. 3). In addition to increased de novo glutamine synthesis, we also found higher levels of intracellular $^{13}C_5$-glutamine in both HCC1806 and MDA-MB-231 A2KO cells after 24 h (Fig. 3A). This is in stark contrast to the low levels of glutamine uptake detected in A2KO cells after 30 min of incubation with radiolabelled glutamine tracer (Fig. 1F). This discrepancy could be due to a reduced rate of glutamine transport caused by the loss of ASCT2, which while evident at 30 min, is compensated for by 24 h. Multiple mechanisms could be contributing to intracellular glutamine levels, including de novo glutamine synthesis and catabolism (Fig. 3), reduced efflux of glutamine (Fig. 4E, SLC7A5 downregulation), and induction of macropinocytosis (Fig. 5). However, the Seahorse glutamine injection data suggest that uptake of glutamine to increase OCR occurs rapidly, further supporting the role of macropinocytosis in this process. It is possible that the discrepancy in the glutamine uptake observed in radiolabelled uptake assays occurs due to a lack of macropinocytosis in suspension cells for this acute assay, compared to Seahorse and metabolomics, which are carried out on adherent cells.

Macropinocytosis is a mechanism utilised by several tumour types to adapt to changing nutrient environments by scavenging nutrients from their environment (Palm, 2019; Finicle et al, 2018; Recouvreux and Commisso, 2017). This process involves non-selective internalisation of extracellular material, including amino acids and proteins, which, after lysosomal fusion and degradation of macropinosomes can replenish the amino acid pool and sustain cell proliferation. While some tumour types exhibit constitutive macropinocytosis driven by RAS or the PI3K pathway (Commisso et al, 2013; Kamphorst et al, 2015; Kim et al, 2018), macro-pinocytosis can also be induced when glutamine (Lee et al, 2019) or amino acids (Jayashankar and Edinger, 2020) are reduced in the media. Previous data have shown that breast cancer cells require more than just macropinocytic protein (e.g. BSA) to support growth (Jayashankar and Edinger, 2020), and it was clear that our A2KO cells similarly replenished glutamine from $^{13}$C-labelled glutamine or glucose (Fig. 3A,I), and not from BSA (Fig. EV5C), nor were A2KO cells more sensitive to chloroquine inhibition (Fig. EV5D).

Upregulation of *SLC38A5*, an amino acid transporter and a positive regulator of macropinocytosis, has been observed in TNBC tumours (Ramachandran et al, 2021). However, in the present study, *SLC38A5* (SNAT5) was not detected in the proteomics data and there was no change in gene expression between WT and the ASCT2 knockout clones (Fig. EV4B), suggesting that SNAT5 is not compensating for the loss of ASCT2. Glutamine deprivation in some pancreatic cancer cell lines induced macropinocytosis via the EGFR signalling pathway (Lee et al, 2019; Lambies et al, 2024). HCC1806 ASCT2 knockout clones showed a trend toward increased EGFR protein expression compared to WT control (Fig. 4C). This suggests that EGFR may play a role in the inducible macropinocytic phenotype observed with the loss of ASCT2 in HCC1806 cells as well. Interestingly, the macropinocytic phenotype (Figs. 5, 6) observed after the loss of ASCT2, varied from constitutive (MDA-MB-231 and MDA-MB-468), inducible (HCC1806) to non-macropinocytic (HCC1569). This variation in macropinocytosis has also been observed after amino acid deprivation in breast cancer cell lines (Jayashankar and Edinger, 2020). Finally, the lack of growth adaptation observed in non-macropinocytic HCC1569 cells on loss of ASCT2 (Fig. 6) suggests that in the other breast cancer cell lines (HCC1806, MDA-MB-231 and MDA-MB-468), macropinocytosis is playing a crucial role in cell proliferation by potentially sustaining the supply of glutamine. These data suggest that while targeting ASCT2 may not be sufficient to impair growth in TNBC, combining it with an inhibitor of macropinocytosis could be an effective treatment. Interestingly, a previous study took advantage of the enhanced macropinocytic ability of TNBC cells to induce an apoptotic response caused by the uptake of verteporfin (Dai et al, 2021), which could be potentially tested in combination with ASCT2 inhibitors.

Overall, these data show that ASCT2-mediated glutamine uptake may not be a bona fide therapeutic target in cancer subtypes that undertake either constitutive or induced macropinocytosis. As has previously been shown for glutaminase inhibitors, the metabolic flexibility of TNBC can provide a variety of avenues for resistance to emerge. While it is clear that the TNBC subset is highly metabolic (Quek et al, 2022), this hardwired metabolic phenotype shows plasticity in response to metabolite restrictions. Future work will need to examine whether there are combinations of metabolic agents that may circumvent these resistance pathways.

# Methods

## Reagents and tools table

| Reagent/Resource | Reference or source | Identifier or catalogue number |
|---|---|---|
| RPMI 1640 Medium, no glutamine | Gibco | Cat #21870076 |
| DMEM, high glucose, pyruvate, no glutamine | Gibco | Cat #10313021 |
| RPMI 1640 Medium, no glucose | Gibco | Cat #11879020 |
| RPMI (glutamine and glucose free) | Elabscience | Cat #PM150125 |
| Sterile foetal bovine serum (FBS) | Bovogen Biologicals, Melbourne, Australia | Cat #SFBS-AU |
| Dialysed FBS | Gibco | Cat #30067-334 |
| TrypLE™ Express | Invitrogen | |
| Phosphate Buffered Saline (PBS) | Gibco | Cat #14190144 |
| Lipofectamine™ LTX and PLUS™ reagents | Invitrogen | Cat #15338100 |
| Tet System Approved FBS | Takara Bio | Cat #631107 |
| BCA Protein Assay kit | Thermo Fisher Scientific | Cat #12605028 |
| NuPAGE™ 4–12% Bis-Tris Protein Gels | Invitrogen | Cat #NP0322 |
| SuperSignal™ West Pico chemiluminescent substrate | Thermo Fisher Scientific | Cat #34580 |
| [3H]-L-glutamine | Perkin Elmer | Cat #NET551 |
| Betaplate scintillation fluid | Perkin Elmer | Cat #1205-440 |
| Fatty acid-free BSA | Sigma | Cat #A8806 |
| Chloroquine | Sigma | Cat #C6628 |
| Crystal Violet | Sigma | Cat #V5265 |
| RPMI XF Medium | Agilent | Cat #103576-100 |
| $^{13}$C$_5$ glutamine | Sigma | Cat #605166 |
| $^{13}$C$_6$ glucose | Sigma | Cat #389374 |
| De-quenched DQ™Green BSA | Sigma | Cat #D12050 |
| Lysotracker Red | Thermo Fisher Scientific | Cat #L7528 |
| TMR-Dextran, 70 kDa | FinaBio | Cat #210 |
| TRIzol™ Reagent | Invitrogen | Cat #15596018 |
| **Experimental Models** | | |
| HCC1806 | American Type Culture Collection (ATCC) | CRL-2335 |

| Reagent/Resource | Reference or source | Identifier or catalogue number |
|---|---|---|
| MDA-MB-231 | ATCC | HTB-26 |
| MDA-MB-468 | ATCC | HTB-132 |
| MCF7 | ATCC | HTB-22 |
| T47D | ATCC | HTB-133 |
| HCC1569 | ATCC | CRL-2330 |
| HEK293 | ATCC | CRL-3216 |
| Female Balb/c nude mice | Animal Resource Centre, Perth, Australia | |
| **Recombinant DNA** | | |
| U6gRNA-pCMV-Cas9-2A-GFP plasmid | Sigma-Aldrich/Merck | N/A |
| Sequence for exon 7 guide | Sigma-Aldrich/Merck | cctcgaagcagtcaacctcccg |
| Sequence for exon 4 guide | Sigma-Aldrich/Merck | agcgccacaccaaagacga |
| pCCL-tet-eGFP-P2A-Luc2 plasmid | Addgene | N/A |
| pCCL-tet-eGFP-P2A-ASCT2 plasmid | This study | N/A |
| pMDLg/prre plasmid | Elim Pharmaceuticals | N/A |
| pRSVRev plasmid | Elim Pharmaceuticals | N/A |
| pMD2.VSV-G plasmid | Elim Pharmaceuticals | N/A |
| **Antibodies** | | |
| ASCT2 antibody (IHC) | Sigma | Cat #HPA035240 |
| ASCT2 antibody (Westerns) | Cell Signaling Technology (CST) | Cat #8057S |
| GAPDH antibody | Abcam | Cat #ab8245 |
| SNX9 antibody | Abcam | Cat #ab181856 |
| SRC antibody | CST | Cat #2108S |
| **Oligonucleotides and other sequence-based reagents** | | |
| *SLC1A5* exon 7 forward primer | This study | 5′ GGCATCTGCCTA ACCTCC 3′ |
| *SLC1A5* exon 7 reverse primer | This study | 5′ ATCCTCTTTCCC AGGGGTA 3′ |
| **Software** | | |
| Morpheus | https://software.broadinstitute.org/morpheus (Broad Institute) | N/A |
| iBright Analysis software | Thermo Fisher Scientific | N/A |
| ImageJ (Fiji) | github.com/fiji/fiji | N/A |
| Integrated Genomics Viewer (IGV) | https://igv.org/ (Broad Institute) | N/A |
| MaxQuant | https://www.maxquant.org/ | N/A |
| Python | https://www.python.org/ | N/A |
| Skyline 22.1 | University of Washington | N/A |

| Reagent/Resource | Reference or source | Identifier or catalogue number |
|---|---|---|
| Wave | Agilent technologies | N/A |
| **Other** | | |
| BD Influx™ | BD | N/A |
| BD FACSAria™ II | BD | N/A |
| LAS 4000 | ImageQuant | N/A |
| iBright FL1500 Imaging System | Thermo Fisher Scientific | N/A |
| IncuCyte® S3 | Sartorius | N/A |
| MicroBeta2 Microplate Counter | Perkin Elmer | N/A |
| Spectramax® 190 Microplate Reader | Molecular Devices | N/A |
| Seahorse XF Analyser | Agilent Technologies | N/A |
| 1260 Infinity LC System | Agilent Technologies | N/A |
| QTRAP6500+ Mass spectrometer | ABSciex | N/A |
| Q Exactive-HFX mass spectrometer | Thermo Fisher Scientific | N/A |
| Zeiss LSM 880 | ZEISS | N/A |

## Immunohistochemistry

The retrospective TNBC patient cohort as previously published (Beckers et al, 2016) was approved by the Royal Prince Alfred Hospital Human Ethics Review Committee (X14-0241; X15-0388). Immunostaining of TMA sections using an ASCT2 rabbit antibody (HPA035240, Sigma) was performed as previously published (Wang et al, 2021) and scored by a pathologist as 0 (negative), 1+ (weak positive), 2+ (moderate positive) or 3+ (strong positive) based on maximum epithelial ASCT2 expression and the percent of TMA stained (0–100%). H-score was calculated as score × %.

## Tissue culture and cell lines

All cells were maintained at 37 °C at 5% $CO_2$ and cultured in 25, 75 or 175 $cm^2$ tissue culture flasks with filter caps (Corning®). Adherent cells were passaged every 2–4 days, as required using, 1× phosphate buffered saline (PBS) (Life Technologies) and TrypLE™ Express (Invitrogen). All cell lines were tested and negative for mycoplasma contamination by staff at the Mycoplasma Testing Facility (UNSW Sydney, Sydney, Australia) using the MycoAlert™ Mycoplasma Detection Kit (Lonza, Basel, Switzerland) every three months.

Cell lines used in this study (Table 1) were purchased from ATCC and revived from low passage authenticated stocks. Cells were cultured in base media specified in Table 1, supplemented with 10% (v/v) sterile foetal bovine serum (FBS) (Bovogen Biologicals, Melbourne, Australia) 1% penicillin/streptomycin solution (Life Technologies) and additional components as specified.

**Table 1. Cell lines and growth media used in this study.**

| | Cell line | Gibco™ base media | Cat # | Glucose | HEPES (mM) | Sodium pyruvate (mM) | Insulin (U/mL) | Gln (mM) |
|---|---|---|---|---|---|---|---|---|
| 1 | HCC1806 | RPMI 1640 Medium, no glutamine | 21870076 | / | / | / | / | 2 |
| 2 | MDA-MB-231 | DMEM, high glucose, pyruvate, no glutamine | 10313021 | / | / | / | / | 2 |
| 3 | MDA-MB-468 | DMEM, high glucose, pyruvate, no glutamine | 10313021 | / | / | / | / | 2 |
| 4 | MCF7 | RPMI 1640 Medium, no glutamine | 21870076 | 11.1 mM (2 g/L) | / | / | / | 4 |
| 5 | T47D | RPMI 1640 Medium, no glucose | 11879020 | 25 mM (4.5 g/L) | 10 | 1 | 0.2 | / |
| 6 | HCC1569 | RPMI 1640 Medium, no glutamine | 21870076 | 11.1 mM (2 g/L) | / | / | / | 2 |
| 7 | HEK293 | DMEM, high glucose, pyruvate, no glutamine | 10313021 | / | / | / | / | 2 |

## Generation of ASCT2 KO cell lines

For ASCT2 knockout, cells were transfected with the plasmid U6gRNA-pCMV-Cas9–2A-GFP containing one of two guides targeting *SLC1A5* (cctcgaagcagtcaacctcccg targeting exon 7 for single-cell clones K1–K6 and all polyclonal lines denoted A2KO; agcgccacac-caaagacga targeting exon 4 for the polyclonal line denoted A2KO#2, Sigma-Aldrich). Negative control lines 'NC' and 'NC#2' were generated with U6gRNA-pCMV-Cas9–2A-GFP containing the guide tatgtgcggcaaaccaagcg (CRISPR08; Sigma-Aldrich/Merck). For three days prior to transfection, culture conditions were as described above, except K1–K6 were cultured with dialysed FBS (NZ Origin, cat# 30067-334, Life Technologies) and varied glutamine concentration (2 mM for K2 and K5, and 0.5 mM for K1, K3, K4 and K6). Cells were then transfected with Lipofectamine™ LTX and PLUS™ reagents (Invitrogen™), incubated in Opti-MEM™ Reduced Serum Medium (Gibco™) for 4 h and subsequently cultured in their original media for 48 h. GFP-expressing cells were then isolated via flow cytometry (FACS; BD Influx™ or BD FACSAria™ II) and grown as single colonies for up to four weeks in original glutamine concentrations before expansion in normal growth media (K1–K6), or as polyclonal populations for up to two weeks in normal growth media (NC, NC#2, A2KO and A2KO#2). Following expansion, GFP-positive polyclonal lines underwent FACS sorting for ASCT2 expression. Cells were trypsinised and resuspended in FACS buffer (1× Ca²⁺ and Mg²⁺ free PBS, 2 mM ethylenediaminetetraacetic acid (Life Technologies), 2% FBS Bovogen), blocked with 0.5% BSA for 20 min, incubated with primary antibody (anti-human ASCT2 from MedImmune diluted 1:3000 in FACS buffer) for 30 min and secondary antibody for 20 min (Goat anti-human IgG Fc-PE secondary antibody (Invitrogen) diluted 1:250 in FACS buffer). Cells then underwent FACS sorting BD Influx™ cell sorted or a BD FACSAria™ II cell sorter. PE-positive (ASCT2 positive) and negative cells were collected for NC and A2KO lines, respectively. One single-cell HCC1806 KO cell line (Bröer et al, 2019) was a kind gift from Prof. Stefan Broer (ANU, Canberra, Australia). Knockouts were verified by Western blot for ASCT2.

To achieve stable expression of ASCT constructs in breast cancer cells, the same methods described for the generation of single-cell HCC1806 ASCT2 KO cell lines were again used for transfection, transduction and FACS sorting, with the following exceptions. Firstly, the target plasmids were the pCCL-tet-eGFP-P2A-ASCT2 (A2) constructs and pCCL-tet-eGFP-P2A-Luc2

(empty vector, EV, as a control). Secondly, the packaging plasmids were pMDLg/prre, pRSVRev and pMD2.VSV-G (Elim Pharmaceuticals), mixed in the microgram ratio 6.5:2.5:3.5. Pre-mixed packaging plasmids (12.5 µg) were added to each lentiviral target mix, along with 8 µg of each target plasmid. The transduced cell lines generated were referred to as HCC1806 WT + EV, HCC1806 KO + EV and HCC1806 KO + A2.

## Breast cancer xenograft model

Female Balb/c nude mice (Animal Resource Centre, Perth, Australia) at 6–8 weeks of age were housed at UNSW Lowy Cancer Centre mouse facility in accordance with UNSW animal ethics guidelines (Approval: 18/61 A). Mice were anesthetised by 5% and sustained at 2% isoflurane inhalation and $2 \times 10^6$ HCC1806 WT or K2 cells in 100 µL of 1:1 Matrigel:PBS were injected into the left and right lower mammary fat pads. Five mice were injected for each cell line, and mice were monitored and tumours were measured by callipers three times per week for 20 days which corresponded with the ethical endpoint at tumour volume of 1000 mm³ calculated by using the formula Volume = Length × Width² × π/6. Animals were sacrificed at an ethical endpoint and tumours were measured and weighed as done previously (Van Geldermalsen et al, 2016).

## Inducible knockdown of ASCT2

Inducible knockdown of ASCT2 was carried out as described previously (Van Geldermalsen et al, 2016), using the lentiviral vector pFH1t(INSR)UTG vector (Herold et al, 2008) containing the plant microRNA sequence ath-miR159a (shCtrl) or a short hairpin RNA (shRNA) targeting ASCT2 (Van Geldermalsen et al, 2016).

## Western blot analysis

Cells ($1.5–2 \times 10^5$ per well) were cultured in normal growth media for 24 h, or media with Tet System Approved FBS (Takara Bio cat# 631107; Mountain View, USA) with and without doxycycline hyclate for 72 h and lysed in RIPA buffer (Sigma-Aldrich). Equivalent protein samples (Micro BCA™ Protein Assay Kit; Thermo Scientific™) were electrophoresed on SDS polyacrylamide gels (NuPAGE™ 4–12% Bis-Tris Protein Gels; Invitrogen™) and transferred to PVDF membranes (Millipore; Merck). Membranes

were blocked in 2.5% (*w/v*) BSA in TBST and incubated with primary and secondary antibodies. A signal was detected using SuperSignal™ West Pico chemiluminescent substrate (Thermo Scientific™) and imaged using an ImageQuant™ LAS 4000 (IL, USA) or iBright.

## Radiolabelled amino acid uptake assay

Cells were trypsinised from normal culture or doxycycline pre-treatment conditions described above. Cells ($5 \times 10^4$) were loaded into each well of a 96-well plate containing 100 nM [³H]-ʟ-glutamine, incubated at 37 °C for 30 min, aspirated and filtered using a Harvester 96 (TOMTEC, CT, USA). Scintillation counting using scintillation fluid (Betaplate scint, Perkin Elmer®) and a MicroBeta2 Microplate Counter (Perkin Elmer®) was then performed.

## CCK8 and MTT assays

Cells were seeded in triplicate into 96-well plates and allowed to adhere overnight before carrying out any treatments. Cells were treated on the following day for low glutamine experiments in 10% dialysed FBS (Gibco, #30067-334), 5% fatty acid-free BSA (Sigma, #A8806-5G) or 5–50 µM chloroquine (Sigma #C6628, 100 mM stock made in water) and the matched vehicle control. For CCK8 assays (Sigma-Aldrich), CCK8 reagent was added for 2 h and absorbance was measured at 450 nm on a Spectramax® 190 Microplate Reader (Molecular Devices, CA, USA) at timepoints indicated. For MTT assays (Millipore; Merck), MTT reagent was added for 4 h and absorbance measured as 570 nm (cell absorbance) – Absorbance 630 nm (background).

## IncuCyte®

Cells were seeded in triplicate into 96-well plates and imaged in an IncuCyte® S3, at 37 °C/5% $CO_2$ using the phase contrast filter and the 10× objective. For each well, confluence (%) values represent the mean of 16 images (HCC1569) or three images (all others) taken.

## Colony formation assay

Polyclonal NC and A2KO cells for each breast cancer cell line were seeded at low densities in six-well plates ($1 \times 10^3$–$5 \times 10^3$ cells/well) in 2 mL of their respective media. Cells were cultured for 12–14 days, with media changes done twice a week until distinct colonies were observed. At endpoint, colonies were washed with PBS (Thermo Fisher, Cat. 14190144) and fixed with 100% methanol (Sigma-Aldrich, Cat. 34860-1L-R) for 30 min. Fixed colonies were stained with 0.5% crystal violet solution (2.5% methanol-crystal violet (Sigma, Cat. V5265-500ML) at a 50:50 ratio v/v) for 20 min, washed twice with MilliQ water and imaged using the iBright FL1500 Imaging System (Thermo Fisher). Colony number and area were quantified using Fiji (v.1.53q, ImageJ) and normalised to the matched control (NC cells).

## Modified Seahorse XF assay for glutamine and glucose reliance

NC and A2KO cells were seeded into Seahorse XFp Cell plate (HCC1806: $1 \times 10^4$ cells/well; MDA-MB-231: $2 \times 10^4$ cells/well) in

full growth media, rested for 60 min in the tissue culture hood and then incubated overnight at 37 °C/5% $CO_2$. The sensor cartridge was humified overnight as per the manufacturer's instructions. The following day growth media in the Miniplate was replaced by pre-warmed complete RPMI XF medium containing appropriate concentrations of glutamine and glucose, XF medium without glutamine (for glutamine reliance) or XF medium without glucose (for glucose reliance). The cell plate was incubated for 60 min in a non-$CO_2$ incubator at 37 °C before measuring oxygen consumption rate (OCR) and extracellular acidification rate (ECAR) every 6.5 min on the Agilent Seahorse XF Analyser. The rate of glutaminolysis was determined by measuring the OCR, a proxy for mitochondrial respiration. Rate of glycolysis was determined by measuring ECAR caused by lactate production a proxy for glycolysis.

For cells in XF media without glutamine, 2 mM glutamine was injected after the 14 min reading. For cells in XF media without glucose, 11 mM (HCC1806) or 25 mM (MDA-MB-231) glucose was injected after the 14 min reading. For both experiments, 3 mM BPTES (glutaminolysis inhibitor) was injected after the 33 min reading and 50 mM 2-DG (2-deoxyglucose, glycolysis inhibitor) was injected after the 53 min reading. Concurrently on the same plate, a set of wells were incubated in a complete XF medium (with both glutamine and glucose) to determine the baseline OCR and ECAR over the first 14 min. OCR or ECAR baseline data represent the average of the initial three readings before the first injection, while 15 min glutamine or glucose injection data were readings 5 and 6 after stabilisation post-injection. Data were generated from three independent experiments with 6 or 8 technical replicates for each condition. One-way ANOVA with Šídák's multiple comparisons test were used to determine statistical significance between media conditions and NC and A2KO.

## Metabolite extraction for glutamine and glucose tracing

Polyclonal HCC1806 NC and A2KO cells were seeded in duplicate at $0.5 \times 10^6$ cells per well, and MDA-MB-231 NC and A2KO cells were seeded at $0.7 \times 10^6$ cells per well in a six-well plate (Thermo) in complete media and allowed to adhere overnight. Cells were then washed with PBS and incubated with glutamine/glucose-free media containing 10% dialysed FBS (Gibco, #30067-334) and either 2 mM $^{13}C_5$-glutamine (Sigma) or $^{13}C_6$-glucose (Sigma) at 11.1 mM for HCC1806 in RPMI or 25 mM MDA-MB-231 in DMEM for 24 h. After 24 h, media was collected from each well, cells were rinsed and metabolites extracted using methanol:acetonitrile:water (5:3:2 (v/v) proportion for glutamine tracing and 2:2:1 (v) proportion for glucose tracing) as described previously (Wang et al, 2021; Quek et al, 2022). Cell pellets from each sample were used for protein quantification by the Lowry assay as described previously (Lowry OH, 1951).

## Targeted metabolomics for glutamine and glucose tracing

Targeted metabolomics for glutamine tracing was carried out as described previously using the LC-MS system (Wang et al, 2021). For glucose tracing, dehydrated samples were reconstituted in a buffer consisting of 20 mM ammonium hydroxide, 20 mM ammonium acetate and 5% acetonitrile before being aliquoted.

Metabolites from the TCA cycle and glycolysis were detected using the 1260 Infinity LC System (Agilent) coupled to the QTRAP6500+ (ABSciex) mass spectrometer. LC separation was performed on a Synergi Hydro-RP LC column (2.5 μm, 2.0 mm × 100 mm, Phenomenex). The buffers and gradient used for LC separation have been described previously (Lu et al, 2010). Skyline 22.1 (University of Washington) was used for chromatographic alignment, peak identification and integration before data were exported for further analysis. The total peak area of metabolites detected by both LC-MS platforms was normalised to the protein concentration of cell pellets collected from the metabolite extraction of lysates.

## Cell lysate preparation for proteomics

HCC1806 WT and KO cells were seeded in 10 cm plates and lysed at >80% confluency using 500 μL denaturing lysis buffer (4% SDS, 20 mM sodium phosphate pH 6.0, 100 mM NaCl, Complete Protease Inhibitors (EDTA-free) and phosphatase inhibitors with scraping to collect the lysates. The extracts were heated at 65 °C for 10 min and sonicated using the QSonica Q800R2 30 s ON, 30 s OFF for 10 min total at 80% amplitude at room temperature. The extracts were centrifuged at 18,000×g for 10 min at 18 °C, and supernatants stored at −20 °C till further processing for proteomics.

## Proteomics using mass spectrometry

Lysate proteins from each sample (100 μg) were precipitated for proteome analysis using chloroform-methanol precipitation as described previously (Harney et al, 2021). The precipitated protein was dissolved in 8 M urea and 0.1 M Tris-HCl pH 8.0 at room temperature by vortexing, then diluted to 1 M urea. Trypsin (1 μg) was added and proteins were digested for 16 h at 37 °C. Trifluoroacetic acid was added to 1% final concentration and 10 μg of peptides were purified using SDB-RPS StageTips as described previously (Harney et al, 2019). The digested peptides were separated by nano-LC using a Thermo Scientific Dionex Ultimate 3000 UHPLC. Samples (injection volume 3 μL–1 μg total peptide) were directly injected into a 50 cm × 75 μm C18 column (Dr. Maisch, Ammerbuch, Germany, 1.9 μm). Peptides were eluted using a linear gradient from 5% of buffer B (80% acetonitrile and 10 mM ammonium formate) to 30% of buffer B for the first 125 min, then 60% of buffer B for the next 5 min and 98% for the remaining time at a flow rate of 500 nL/min over a total run time of 140 min. Peptides were ionised by electrospray ionisation at 2.3 kV. Tandem mass spectrometry analysis was carried out on a Q Exactive-HFX mass spectrometer (Thermo Fisher) using HCD fragmentation in positive mode. The data-dependent acquisition method used acquired MS/MS spectra of the top 20 most abundant ions at any one point during the gradient. MS1 scans were acquired from 350–1400 $m/z$ (60,000 resolution, $3 \times 10^6$ AGC target, 20 ms maximum injection time), and MS2 scans having a fixed first $m/z$ of 140 (15,000 resolution, $1 \times 10^5$ AGC target, 25 ms maximum injection time, 27 NCE, 1.4 $m/z$ isolation width). RAW data files were analysed using the integrated quantitative proteomics software and search engine MaxQuant. A false discovery rate (FDR) of 1% using a target-decoy-based strategy was used for protein and peptide identification. The database used for identification contained the Uniprot human database alongside the MaxQuant contaminants database. Mass tolerance was set to 4.5 ppm for precursor ions and 20 ppm for fragments. Trypsin was set as the digestion enzyme with a maximum of two missed cleavages. Oxidation of Met, deamidation of Asn/Gln, pyro-Glu/Gln and protein N-terminal acetylation were set as variable modifications. Carbamidomethylation of Cys was set as a fixed modification. The MaxLFQ algorithm was used for label-free quantitation.

## RNA Sequencing (RNA-Seq)

RNA was extracted from $1 \times 10^6$ cells using TRIzol™ Reagent (Invitrogen™). Two biological replicates of each sample (>20 μL, >50 ng/μL, 260/280 > 2) were then sequenced at NovogeneAIT Genomics Singapore Pte Ltd by paired-end sequencing. The sequencing reads were mapped to the human genome, GRCh38.p13, available from Ensembl 101 (Yates et al, 2020). Mapping and counting of reads were carried out by running modified STAR scripts (Dobin et al, 2013) in the computational cluster Katana (UNSW Sydney, Sydney). Log$_2$-mean expression (as indicated) of genes within conditions and log$_2$-fold changes in gene expression between conditions are presented. Datapoints represent biological replicates. Data were analysed using Python libraries such as pandas and visualised using Morpheus (https://software.broadinstitute.org/morpheus).

## Dextran uptake and co-localisation assay

Cells ($0.05 \times 10^6$/well for dox-treated cells and corresponding controls, $0.2 \times 10^6$/well for other cells) were seeded on glass coverslips (Menzel Gläser, Thermo Scientific) in a 24-well plate and cultured for 72 h (dox-treated cells and corresponding controls) or overnight (all other cells). Dextran uptake, imaging and analysis procedures were adapted from the protocol described in (Commisso et al, 2014). Cells were incubated in media containing 1 mg/mL TMR-Dextran (Tetramethylrhodamine, 70,000 MW, Lysine Fixable) for 30 min, fixed for 30 min with 4% formaldehyde (Thermo Scientific Pierce), and stained with Hoechst stain 1:1000. For co-localisation assays, cells were incubated with 10 μg/mL de-quenched DQ™ Green BSA (Thermo #D12050) and 1:10000 Lysotracker Red (Thermo #L7528) for 1 h and then cells were fixed for 30 min with 4% formaldehyde (Thermo Scientific Pierce) and stained with Hoechst stain 1:1000. Images were collected using a Zeiss LSM 880 (UNSW Sydney) with a 63× oil objective, and analysed in ImageJ (Fiji) (Schindelin et al, 2012; Schindelin et al, 2015) as described in Commisso et al, (2014) to determine the 'total macropinosome area' per field of vision. We report the macropinocytic index as the total macropinosome area divided by the total cell number. The area of co-localisation for DQ-BSA-Green and Lysotracker Red was determined using the BIOP JaCoP image analysis plugin in (Fiji is Just) ImageJ.

## Graphics

BioRender was used by Kanu Wahi to create the synopsis figure (BioRender.com/w02x460), Fig. 1D (BioRender.com/q79r050), Fig. EV1N (BioRender.com/q79r050) and Fig. EV3A (BioRender.com/w02x460).

# Data availability

RNA-seq data: Gene Expression Omnibus GSE256218. https://www.ncbi.nlm.nih.gov/geo/query/acc.cgi?acc=GSE256218. Proteomics data: EBI PRIME accession PXD053894. https://www.ebi.ac.uk/pride/archive/projects/PXD053894/.

The source data of this paper are collected in the following database record: biostudies:S-SCDT-10_1038-S44318-024-00271-6.

# Peer review information

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

## Acknowledgements

This work was supported by grants from the National Breast Cancer Foundation (IIRS-23-050 to JH and L-EQ); Cancer Council NSW (RG18-06 to JH); Tour de Cure (RSP-101-FY2023 to JH) and the Australian Cancer Research Foundation (Tumour Metabolism Laboratory to JH). Flow sorting was performed by Dr Emma Johansson Beves at the UNSW Flow Cytometry facility. Metabolomics and proteomics were facilitated by access to Sydney Mass Spectrometry, a core research facility at the University of Sydney and the Bioanalytical Mass Spectrometry Facility within the Mark Wainwright Analytical Centre of the University of New South Wales. We are grateful to the School of Biomedical Sciences, UNSW for providing access to the premium BioRender licence. We thank Aimee Edinger and Vaishali Jayashankar from the University of California,

Irvine for helpful discussions regarding macropinocytosis and Stefan Broer for the provision of the original ASCT2 knockout line (KO).

## Author contributions

**Kanu Wahi**: Conceptualisation; Data curation; Software; Formal analysis; Supervision; Validation; Investigation; Visualisation; Methodology; Writing—original draft; Writing—review and editing. **Natasha Freidman**: Conceptualisation; Data curation; Formal analysis; Validation; Investigation; Visualisation; Methodology; Writing—original draft; Writing—review and editing. **Qian Wang**: Conceptualisation; Data curation; Formal analysis; Investigation; Methodology; Writing—review and editing. **Michelle Devadason**: Data curation; Formal analysis; Investigation; Writing—review and editing. **Lake-Ee Quek**: Data curation; Formal analysis; Supervision; Investigation; Methodology; Writing—review and editing. **Angel Pang**: Investigation. **Larissa Lloyd**: Data curation; Formal analysis; Investigation; Visualisation. **Mark Larance**: Data curation; Software; Formal analysis; Investigation; Methodology; Writing—review and editing. **Fabio Zanini**: Software; Supervision; Methodology. **Kate Harvey**: Formal analysis; Investigation; Methodology. **Sandra O'Toole**: Formal analysis; Investigation; Methodology. **Yi Fang Guan**: Formal analysis; Investigation; Methodology; Writing—review and editing. **Jeff Holst**: Conceptualisation; Resources; Data curation; Formal analysis; Supervision; Funding acquisition; Visualisation; Methodology; Writing—original draft; Project administration; Writing—review and editing.

Source data underlying figure panels in this paper may have individual authorship assigned. Where available, figure panel/source data authorship is listed in the following database record: biostudies:S-SCDT-10_1038-S44318-024-00271-6.

## Disclosure and competing interests statement

The authors declare no competing interests.

# Expanded View Figures

**Figure EV1.  ASCT2 knockout in additional breast cancer cell lines, relates to Fig. 1.**                                            ▶

(A) MTT assays for polyclonal HCC1806 NC and A2KO cells cultured in various glutamine concentrations with dialysed FBS (dFBS; 10% v/v). Mean ± SEM from three independent experiments performed in triplicate and analysed by two-way ANOVA; ns is not significant. (B–M) Polyclonal MDA-MB-468 (B–E), MCF7 (F–I) and T47D (J–M) NC and A2KO cell lines were generated by CRISPR. (B, F, J) Western blot for ASCT2 protein (CST #8057S, 60–80 kDa), with GAPDH protein (Abcam #ab8245, 37 kDa) as a loading control. (C, G, K) Uptake of 100 nM [$^3$H]-L-glutamine over 30 min, mean ± SEM from three independent experiments in triplicate and analysed by unpaired $T$-test where ****$p < 0.0001$. (D, H, L) CCK8 growth assays over 96 h, measured at timepoints indicated and normalised to the day 0 reading. Cells were seeded at a density of $5 \times 10^3$ (MDA-MB-468) and $1 \times 10^4$ (MCF7 and T47D) per well. Mean ± SEM from three independent experiments performed in triplicate and analysed by two-way ANOVA where ns is not significant. (E, I, M) Colony formation assay (CFA) in a six-well plate at $2.5 \times 10^3$ cells (MDA-MB-468) and $5 \times 10^3$ cells (MCF7 and T47D) per well, fixed and stained with 0.5% crystal violet after 12–14 days. Mean ± SEM from three independent experiments in triplicate and analysed by unpaired $T$-test where ****$p < 0.0001$ (E) and ns (I, M). (N) Schematic for *SLC1A5* gene (ASCT2) and CRISPR guide RNA (red) targeting exon 4. The image is not to scale. Polyclonal HCC1806 NC#2 and A2KO#2 cells generated with exon 4 guide RNA and assessed for (O) ASCT2 expression with GAPDH as control by western blot, (P) [$^3$H]-L-glutamine uptake over 30 min, mean ± SEM from three independent experiments in triplicate and analysed by unpaired $T$-test where ****$p < 0.0001$ and (Q) cell growth (relative to day 0) over 96 h by CCK8 assay, data are mean ± SEM from three independent experiments performed in triplicate and analysed by two-way ANOVA where ns is not significant.

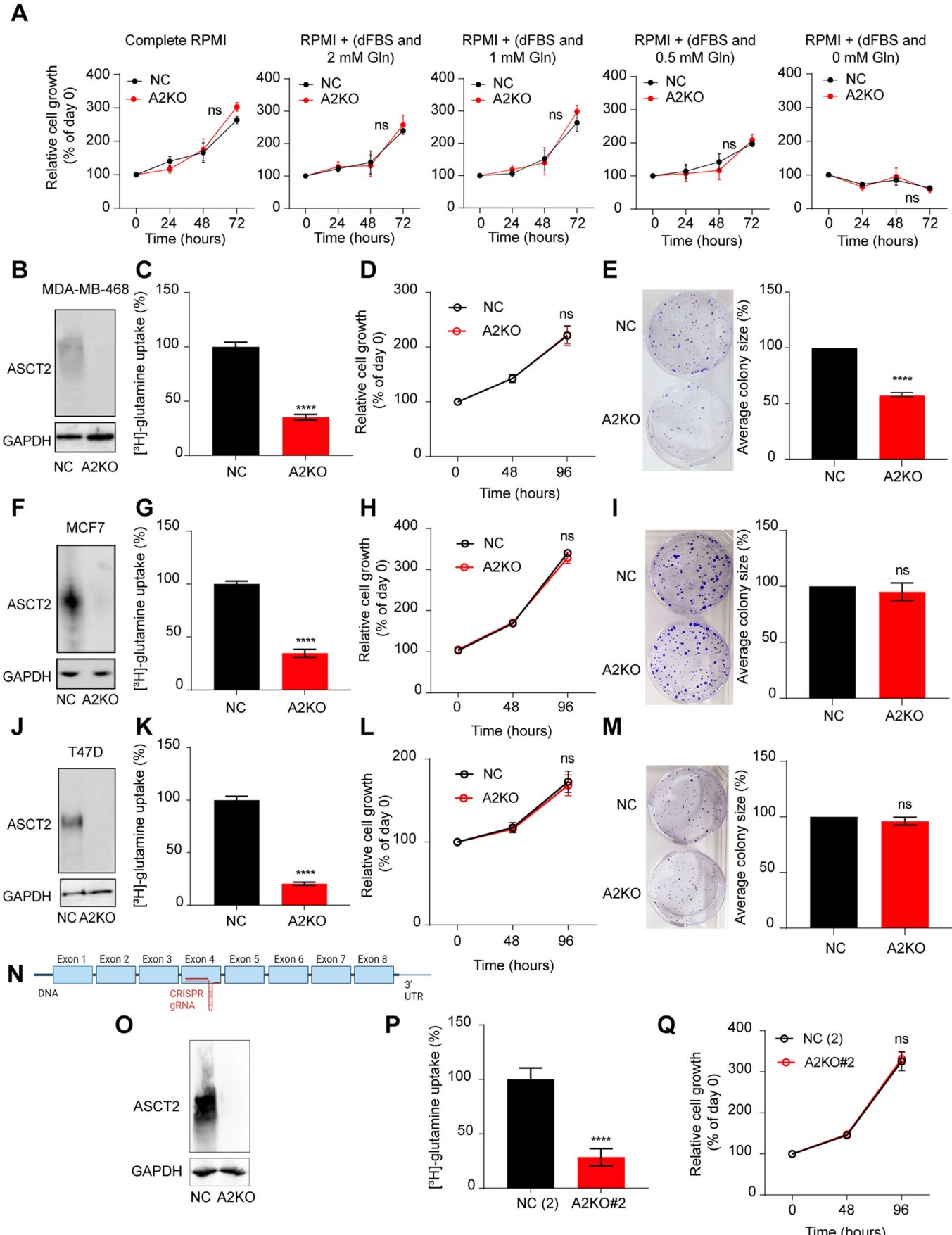

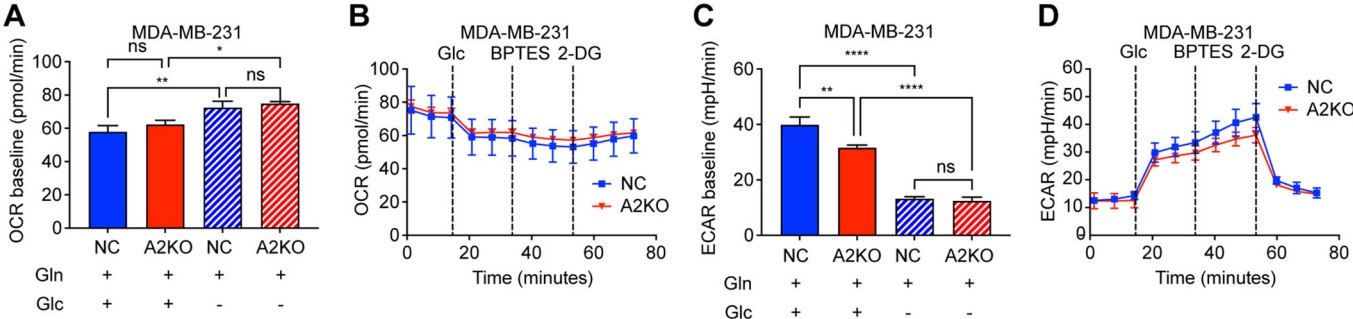

**Figure EV2.  Rate of glycolysis appears to be lower in MDA-MD-231 A2KO cells, relates to Fig. 2.**

(**A**) OCR was measured in MDA-MB-231 cells over 14 min in complete media or media lacking glucose, \*\*$p = 0.0067$, \*$p = 0.0230$. (**B**) Seahorse assay was carried out in MDA-MB-231 cells, and OCR was measured in glucose-free media, with sequential acute injection of 25 mM glucose, followed by 3 µM BPTES (glutaminase inhibitor) and finally 50 mM 2-DG (glycolysis inhibitor). (**C**) ECAR was measured in MDA-MB-231 cells over 14 min in complete media or media lacking glucose, \*\*$p = 0.0044$, \*\*\*\*$p < 0.0001$. (**D**) Seahorse assay was carried out in MDA-MB-231 cells, and ECAR was measured in glucose-free media, with a sequential acute injection of 25 mM glucose, followed by 3 µM BPTES (glutaminase inhibitor) and finally 50 mM 2-DG (glycolysis inhibitor). Data are mean ± SEM from three independent experiments in triplicate, analysed by one-way ANOVA with Šídák's multiple comparisons test (ns not significant).

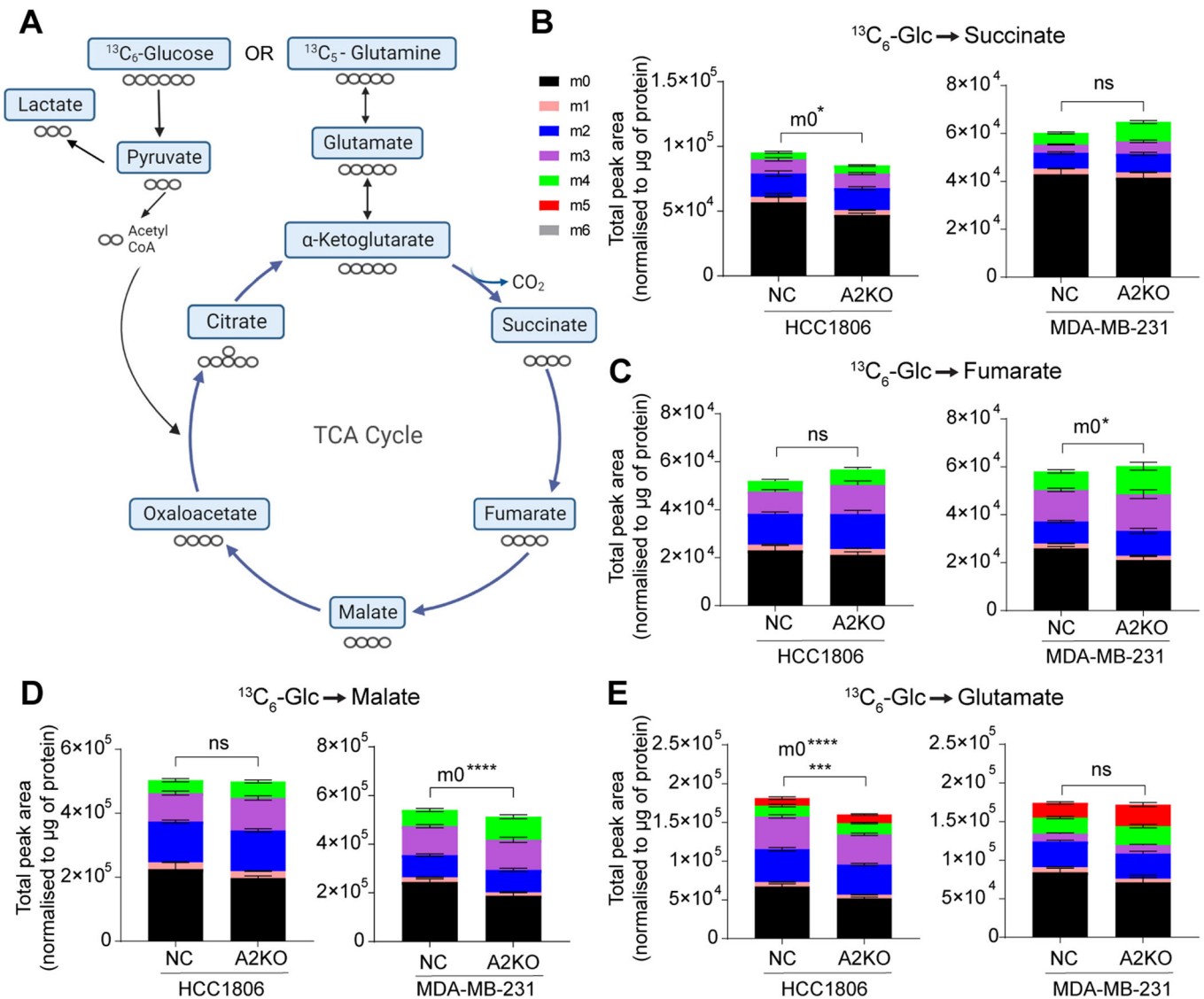

**Figure EV3. TCA metabolite levels remain predominantly unchanged in A2KO cells, relates to Fig. 3.**

(A) Schematic for tracing $^{13}C_5$-glutamine or $^{13}C_6$-glucose carbons through glycolysis and the TCA cycle in polyclonal HCC1806 and MDA-MB-231 NC and A2KO cells. (B–E) Total peak area of $^{13}C_6$-glucose ($^{13}C_6$-Glc) derived TCA metabolites succinate (B), fumarate (C) and malate (D) as well as glutamate (E) were detected by LC-MS at 24 h. *p* values for differences between NC and A2KO cells for (B) succinate in HCC1806 m0 *$p = 0.0344$ and MDA-MB-231 not significant (ns) for all, (D) fumarate in HCC1806 ns for all and MDA-MB-231 m0 *$p = 0.0111$, (D) malate in HCC1806 ns for all and MDA-MB-231 m0 ****$p < 0.0001$, (E) glutamate in HCC1806 ***$p = 0.0006$, m0 ****$p < 0.0001$ and MDA-MB-231 ns for all. Mass of the unlabelled metabolite ($^{12}C = m0$), which changes with integration with $^{13}C$-labelled carbons, where (m#) indicates a metabolite with # of carbons labelled with $^{13}C$. Isotopologue (m0) denotes that all carbons of the metabolite are $^{12}C$, and the metabolite is unlabelled. Isotopologue (m6) signifies that six carbons are $^{13}C$ isotopes ($^{13}C_6$-Glc). The total peak area was normalised to µg of protein for each sample. Error bars are mean ± SEM from three independent experiments performed in duplicate analysed by two-way ANOVA. Where isotopologues are listed with asterisks, the unlisted isotopologues are ns.

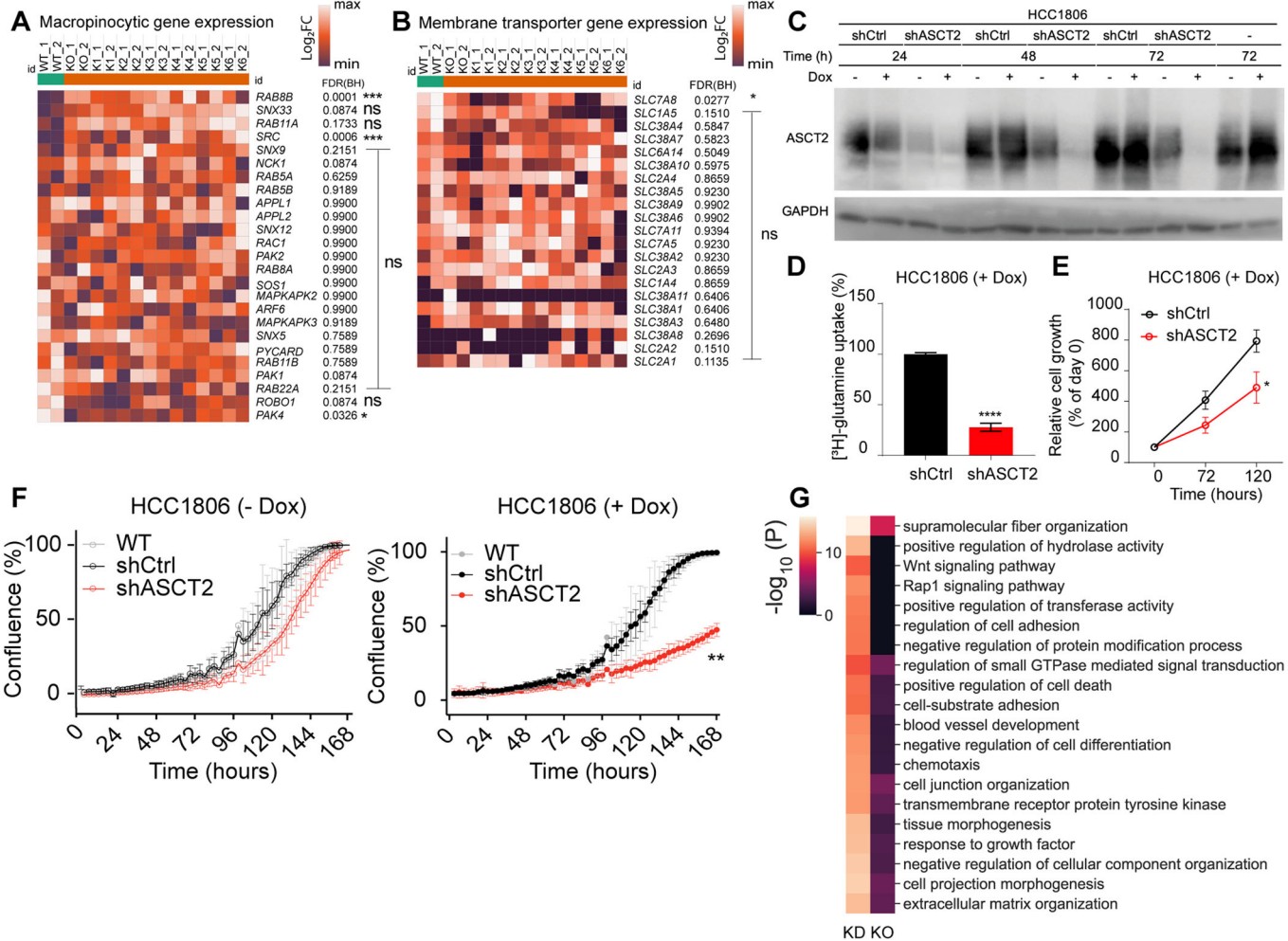

**Figure EV4. Macropinocytic and transporter mRNA expression heatmaps and effects of ASCT2 shRNA knockdown in HCC1806, relates to Fig. 4.**

(A, B) Differentially expressed genes from RNA-seq relevant to macropinocytosis and membrane transport in HCC1806 cells. All data were normalised and log2-transformed, and a pseudocount of 0.1 was applied in DESeq2 v1.26.0 software. DESeq2 was also used to select differentially expressed genes ('DEGs') HCC1806 WT and each HCC1806 ASCT2 knockout clones (KO and K1–K6) ($n = 2$ biological replicates). A relative colour scheme is used for heatmaps (A, C, E, F) where minimum (dark brown) and maximum (beige) values in each row are converted to colours (https://software.broadinstitute.org/morpheus/). (C) Western blot of ASCT2 protein (CST #8057S, 60–80 kDa) and GAPDH protein (Abcam #ab8245, 37 kDa) as a loading control in HCC1806 cell lines stably transfected with shCtrl and shASCT2 after 72 h pre-incubation in dox. Cells were maintained in tetracycline-free media ±1 μg/mL dox (as indicated by -/+ dox). (D) Uptake of 100 nM [3H]-glutamine in HCC1806 cell lines over 30 min, after 72 h pre-incubation in dox, mean ± SEM from three independent experiments in triplicate and analysed by unpaired $t$-test where ****$p < 0.0001$. (E) CCK8 assay of HCC1806 cell lines measured at timepoints indicated ± dox. Cells were seeded at a density of $1 \times 10^3$ cells per well and grown in dox for the duration of the CCK8 assay where mean ± SEM from three independent experiments performed in triplicate and analysed by two-way ANOVA where *$p = 0.0428$. (F) Growth of HCC1806 cell lines measured by live cell imaging (IncuCyte® S3) over 168 h, at 3 h intervals. Cells were seeded at a density of $1 \times 10^3$ cells per well and grown ± dox for the duration of the assay, data are mean ± SEM from three independent experiments performed in triplicate and analysed by two-way ANOVA where **$p = 0.0034$. (G) Heatmap of concordantly enriched pathways comparing ASCT2 knockdown (KD) and ASCT2 KO (KO), where P indicates hypergeometric p value for enrichment computed in Metascape.

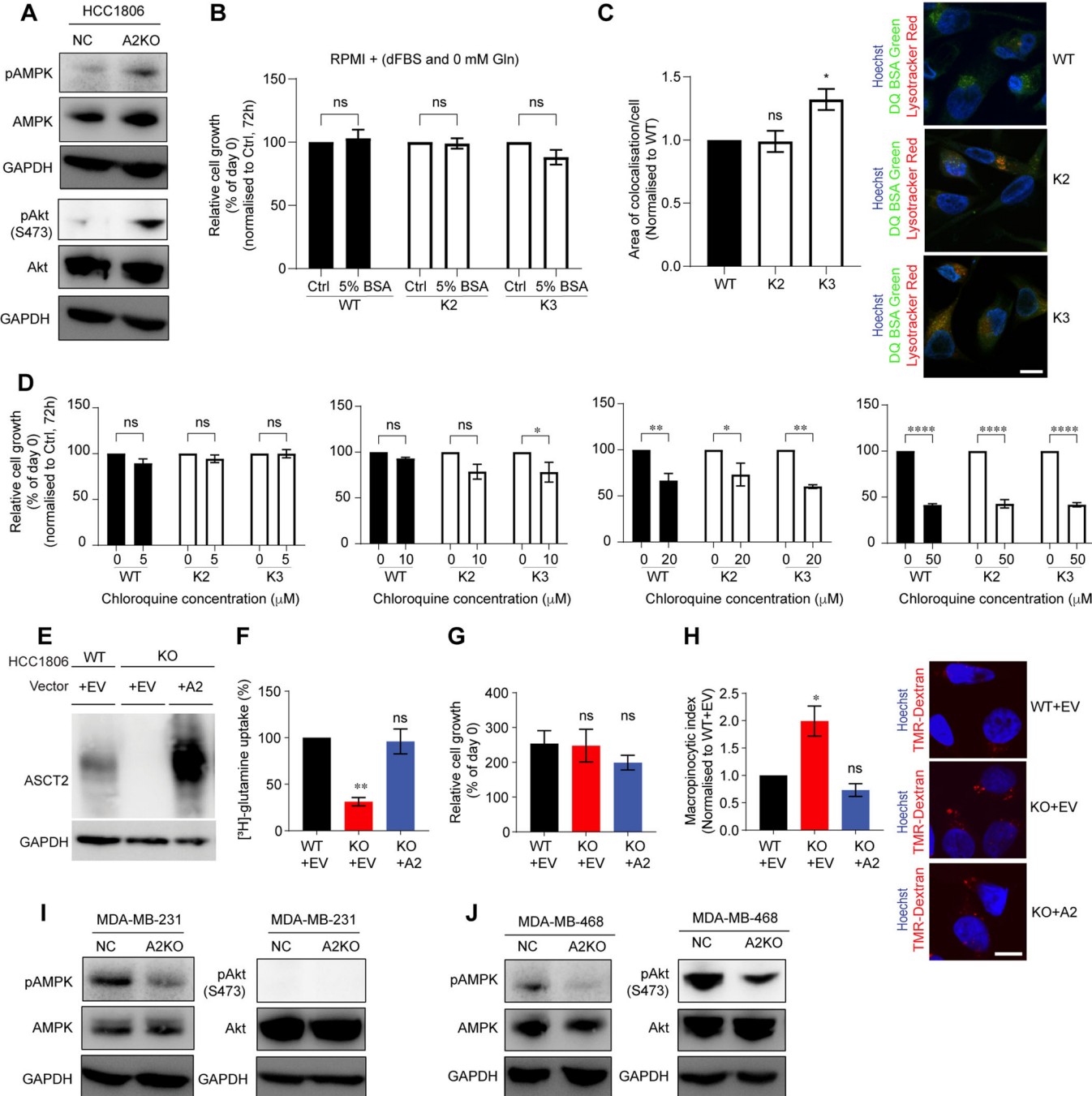

**Figure EV5.** **AMPK and Akt phosphorylation is higher in HCC1806 A2KO cells, relates to Fig. 5.**

(**A**) Western blot of pAMPK protein (CST #2535, 62 kDa), AMPK (CST #5831, 62 kDa) and GAPDH protein (Abcam #ab8245, 37 kDa) and pAKT (S473) protein (CST #9271, 60 kDa), Akt (CST #9272, 60 kDa) and GAPDH protein (Abcam #ab8245, 37 kDa) as a loading control in polyclonal HCC1806 NC and A2KO cell lines. Representative image from two to three independent repeats is shown. (**B**) CCK8 assays for HCC1806 WT, K2 and K3 ASCT2 KO cell lines cultured in low glutamine conditions with dialysed FBS (dFBS; 10% v/v) ±5% fatty acid-free BSA. Mean ± SEM from four independent experiments in triplicate and analysed by two-way ANOVA where ns is not significant. (**C**) Area of DQ-BSA Green and Lysotracker Red co-localisation (yellow) quantified per cell in HCC1806 WT, K2 and K3 ASCT2 KO cell lines. Representative images, from three independent repeats of DQ-BSA Green and Lysotracker Red co-localisation, where nuclei were visualised with Hoescht fluorescent stain (blue). The scale bar in white is equivalent to 8 µM. Mean ± SEM of indices calculated from >30 cells (3–4 fields of view) across three independent experiments. Asterisks indicate $p$ values, ns not significant, $*p = 0.03$ from a one-way ANOVA. (**D**) CCK8 assays for HCC1806 WT, K2 and K3 ASCT2 KO cell lines cultured in 2 mM glutamine with dialysed FBS (dFBS; 10% v/v) ± water control/chloroquine (CQ) (5–50 µM). Mean ± SEM from three independent experiments in triplicate and analysed by two-way ANOVA where ns is not significant, 10 µM CQ $*p = 0.0487$ for K3, 20 µM CQ $**p = 0.0059$ for WT, $*p = 0.0239$ for K2, $**p = 0.0015$ for K3 and 50 µM CQ $****p < 0.0001$ for WT, K2 and K3. (**E**) Western blot of ASCT2 protein (CST #8057S, 60–80 kDa) and GAPDH protein (Abcam #ab8245, 37 kDa) as a loading control in HCC1806 WT cell line transduced with empty vector (EV) or ASCT2 KO cells transduced with EV or ASCT2 (A2; rescue). (**F**) Uptake of 100 nM [³H]-L-glutamine in HCC1806 cell lines over 30 min. Mean ± SEM from three independent experiments in triplicate where asterisks indicate $p$ value from a one-way ANOVA where $**p = 0.002$ and ns is not significant. (**G**) CCK8 assay of HCC1806 cell lines measured at 96 h, mean ± SEM from three independent experiments done in triplicate and analysed by two-way ANOVA where ns is not significant. (**H**) Uptake of TMR-dextran (red) in HCC1806 WT + EV, KO + EV and KO + A2. Nuclei were visualised with Hoescht fluorescent stain (blue). The scale bar in white is equivalent to 10 µM. Quantification of macropinocytic index in HCC1806 cell lines is macropinocytosis/cell (area of macropinosomes/number of cells in field). Representative images are shown from three independent repeats. Mean ± SEM of indices calculated from >30 cells (3–4 fields of view) across three independent experiments. Asterisks indicate $p$ value from a one-way ANOVA where $*p = 0.01$ and ns is not significant). (**I, J**) Western blot of pAMPK protein (CST #2535, 62 kDa), AMPK (CST #5831, 62 kDa) and GAPDH protein (Abcam #ab8245, 37 kDa) and pAKT (S473) protein (CST #9271, 60 kDa), Akt (CST #9272, 60 kDa) and GAPDH protein (Abcam #ab8245, 37 kDa) as a loading control in polyclonal MDA-MB-231 (**I**) and MDA-MB-468 (**J**) NC and A2KO cell lines. Representative image from two to three independent repeats is shown.

