## [Peer Review File · The EMBO Journal]

Macropinocytosis mediates resistance to loss of glutamine transport in triple-negative breast cancer

Kanu Wahi, Natasha Freidman, Qian Wang, Michelle Devadason, Lake-Ee Quek, Angel Pang, Larissa Lloyd, Mark Larance, Fabio Zanini, Kate Harvey, Sandra O'Toole, Yi Fang Guan, and Jeff Holst

Corresponding author(s): Jeff Holst (j.holst@unsw.edu.au) , Kanu Wahi (k.wahi@unsw.edu.au)

Review Timeline:

Submission Date:	26th Feb 24
Editorial Decision:	17th Apr 24
Revision Received:	29th Jul 24
Editorial Decision:	20th Aug 24
Revision Received:	16th Sep 24
Accepted:	23rd Sep 24

Editor: Daniel Klimmeck

Transaction Report:

Dear Dr Jeff Holst,

Thank you for submitting your manuscript for consideration by the EMBO Journal, as well as for your patience with our feedback at this time of the year. Your work has now been seen by two referees with expertise in cancer metabolism and glutamine biology and whose comments are shown below.

Given the overall interest stated and broader angle of your findings, we are able to invite you to revise your manuscript experimentally to address the referees' comments. I need to stress though that we do require strong support from the referees on a revised version of the study in order to move on to publication of the work.

I would appreciate if you could contact me during the next weeks for exchange e.g. a video call to discuss your perspective on the comments and potential plan for revisions.

Please feel free to contact me if you have any questions or need further input on the referee comments.

When submitting your revised manuscript, please carefully review the instructions below.

Please feel free to approach me any time should you have additional questions related to this.

Thank you for the opportunity to consider your work for publication.

I look forward to your revision.

Best regards,

Daniel Klimmeck

Daniel Klimmeck, PhD
Senior Editor
The EMBO Journal

Instruction for the preparation of your revised manuscript:

- 1) a .docx formatted version of the manuscript text (including legends for main figures, EV figures and tables). Please make sure that the changes are highlighted to be clearly visible.
- 2) individual production quality figure files as .eps, .tif, .jpg (one file per figure).
- 3) a .docx formatted letter INCLUDING the reviewers' reports and your detailed point-by-point response to their comments. As part of the EMBO Press transparent editorial process, the point-by-point response is part of the Review Process File (RPF), which will be published alongside your paper.
- 4) a complete author checklist, which you can download from our author guidelines ([https://wol-prod-cdn.literatumonline.com/pb-assets/embo-site/Author Checklist%20-%20EMBO%20J-1561436015657.xlsx](https://wol-prod-cdn.literatumonline.com/pb-assets/embo-site/Author%20Checklist%20-%20EMBO%20J-1561436015657.xlsx)). Please insert information in the checklist that is also reflected in the manuscript. The completed author checklist will also be part of the RPF.
- 5) Please note that all corresponding authors are required to supply an ORCID ID for their name upon submission of a revised manuscript.
- 6) It is mandatory to include a 'Data Availability' section after the Materials and Methods. Before submitting your revision, primary datasets produced in this study need to be deposited in an appropriate public database, and the accession numbers and database listed under 'Data Availability'. Please remember to provide a reviewer password if the datasets are not yet public (see

<https://www.embopress.org/page/journal/14602075/authorguide#datadeposition>).

7) Our journal encourages inclusion of *data citations in the reference list* to directly cite datasets that were re-used and obtained from public databases. Data citations in the article text are distinct from normal bibliographical citations and should directly link to the database records from which the data can be accessed. In the main text, data citations are formatted as follows: "Data ref: Smith et al, 2001" or "Data ref: NCBI Sequence Read Archive PRJNA342805, 2017". In the Reference list, data citations must be labeled with "[DATASET]". A data reference must provide the database name, accession number/identifiers and a resolvable link to the landing page from which the data can be accessed at the end of the reference. Further instructions are available at .

8) At EMBO Press we ask authors to provide source data for the main and EV figures. Our source data coordinator will contact you to discuss which figure panels we would need source data for and will also provide you with helpful tips on how to upload and organize the files.

Numerical data can be provided as individual .xls or .csv files (including a tab describing the data). For 'blots' or microscopy, uncropped images should be submitted (using a zip archive or a single pdf per main figure if multiple images need to be supplied for one panel). Additional information on source data and instruction on how to label the files are available at .

9) We replaced Supplementary Information with Expanded View (EV) Figures and Tables that are collapsible/expandable online (see examples in <https://www.embopress.org/doi/10.15252/emboj.201695874>). A maximum of 5 EV Figures can be typeset. EV Figures should be cited as 'Figure EV1, Figure EV2' etc. in the text and their respective legends should be included in the main text after the legends of regular figures.

11) For data quantification: please specify the name of the statistical test used to generate error bars and P values, the number (n) of independent experiments (specify technical or biological replicates) underlying each data point and the test used to calculate p-values in each figure legend. The figure legends should contain a basic description of n, P and the test applied. Graphs must include a description of the bars and the error bars (s.d., s.e.m.).

We realize that it is difficult to revise to a specific deadline. In the interest of protecting the conceptual advance provided by the work, we recommend a revision within 3 months (16th Jul 2024). Please discuss the revision progress ahead of this time with the editor if you require more time to complete the revisions. Use the link below to submit your revision:

Referee #1:

The manuscript by Wahi K. and colleagues discusses the significance of metabolic reprogramming in supporting cancer cell proliferation, with a particular focus on glutamine addiction in triple-negative breast cancer (TNBC). The study investigates resistance mechanisms, particularly emphasizing macropinocytosis as a potential pathway. It presents experimental data indicating that the complete loss of ASCT2 triggers compensatory mechanisms, such as the upregulation of macropinocytosis in TNBC cells, suggesting caution in targeting glutamine uptake through ASCT2 due to potential resistance pathways. Additionally, the study demonstrates alterations in glutamine and glucose dependencies, glutamine oxidation, and intracellular metabolite levels upon ASCT2 knockout in TNBC cells, providing insights into how cancer cells adapt to metabolic perturbations. Proteomic and mRNA sequencing analyses suggest upregulation of proteins associated with macropinocytosis in ASCT2 knockout cells, further implicating this pathway in resistance mechanisms.

Overall, the study is well conducted, and the conclusions will be relevant to a broad audience. However, there are several points that need to be addressed before publication. Below are my main concerns:

1. In Figure 4, the authors demonstrate increased expression of multiple proteins associated with macropinocytosis, inhibition of ferroptosis, and glutamine efflux in HCC1806 ASCT2 KO cells through whole proteome analysis. However, these results were not validated by a different approach. The expression of selected proteins should be assessed by western blot to solidify these results.
2. The authors conclude that macropinocytosis mediates resistance to the loss of glutamine transport and provide evidence that fluorescently labeled 70 kDa dextran signal is increased in ASCT2 KO cells. However, this evidence may be insufficient for a solid conclusion. This reviewer recommends measuring lysosomal proteolysis using DQ BSA, a self-quenched albumin probe whose fluorescence becomes dequenched upon lysosomal proteolysis in ASCT2 KO cells.
3. Along the same lines, if extracellular material can replenish the glutamine pool and sustain cell proliferation after lysosomal fusion and degradation of macropinosomes, the authors should demonstrate that the addition of BSA to medium without glutamine rescues the proliferative defect in ASCT2 KO cells but not in NC cells.
4. Are ASCT2 KO cells more sensitive to inhibitors of lysosomal activity?

Referee #2:

In the manuscript under review the authors investigate the dependency of triple-negative breast cancer (TNBC) on the plasma membrane glutamine transporter ASCT2 for glutamine uptake to fuel their metabolism and growth. This study is based on their previous identification of ASCT2 as highly upregulated in TNBC. They find that breast cancer cell lines tolerate complete loss of ASCT2 surprisingly well, and deletion of ASCT2 did not result in a decrease in intracellular glutamine levels, and instead were higher than in wild-type cells. The authors trace these metabolic changes back to an upregulation of micropinocytosis, reduction of glutamine efflux through downregulation of SLC7A5, and increased glutamine synthesis in ASCT2 knockout cells, implicating these as a possible resistance mechanisms to inhibiting glutamine uptake through ASCT2 in the context of breast cancer chemotherapy.

By testing whether targeting glutamine uptake by inhibition of ASCT2 could be a viable strategy in the treatment of triple negative breast cancers, the paper seeks to identify new strategies to treat triple negative breast cancers, which in principle is a question of high clinic relevance.

The approach the authors take to directly target ASCT2 is limited in scope but yields some novel insights into the regulation of glutamine homeostasis in breast cancer.

In general, the conclusions of the study are supported by the experiments. However, some major concerns need to be addressed that are essential to support the main conclusions of the study.

Most of the findings rely on ASCT2 KO cells generated by CRISPR/Cas, and the authors target two different exons. They validate loss of ASCT2 protein by western blot.

- As the authors write in the discussion, it is possible that targeting of ASCT2 with a guide targeting exon 7 (and 4 of that matter) results in partial loss of the protein. Thus, it is critical to validate the KO cells appropriately. Next generation sequencing or another sequencing method should be employed to confirm the mutations introduced to the proposed KO cells and determine whether and what part of the coding sequence of ASCT2 is actually deleted in the different clones.
- Because of potential off-target effects of CRISPR, some of the key experiments (such as the proliferation and uptake assays in Figure 1, the macropinocytosis assays in Figure 5 and possibly xenograft studies) should include control cells in which the (guide

refractory) cDNA for ASCT2 is added back to the KO cells. While this is an important control to determine whether observed phenotypes are specific to ASCT2 loss, they might also reveal how flexible the metabolic rewiring in the breast cancer cells is, i.e. whether reexpression of ASCT2 can revert back all phenotypes or whether cells undergo a change that is irreversible or overall metabolically more favorable, which could be essential information for developing ways to target glutamine metabolism in cancer.

Dear Daniel,

Thank you for working with us through the revision process – the teleconference was insightful and a great way to ensure we were on the same page with our planned experiments. We appreciate the thoughtful and balanced reviews from the referees, and have now completed the experiments raised by each of them. We hope you will all agree that the changes have strengthened the manuscript and that it is now acceptable for publication in EMBO J.

We have outlined all the new data and associated manuscript text in this point-by-point response below, which should make it easy for the referees to see the extent of the new data and how it has been incorporated into the manuscript. In addition to the new data below, we have made some text changes to the methods to reflect the new experiments. We have uploaded a tracked changes text version (text additions highlighted in yellow) so that the changes can be simply identified, as well as a clean version ready for publication.

Please do not hesitate to reach out to us if you have any questions.

Kind Regards,
Jeff Holst and Kanu Wahi

Referee #1 comments:

The manuscript by Wahi K. and colleagues discusses the significance of metabolic reprogramming in supporting cancer cell proliferation, with a particular focus on glutamine addiction in triple-negative breast cancer (TNBC). The study investigates resistance mechanisms, particularly emphasizing macropinocytosis as a potential pathway. It presents experimental data indicating that the complete loss of ASCT2 triggers compensatory mechanisms, such as the upregulation of macropinocytosis in TNBC cells, suggesting caution in targeting glutamine uptake through ASCT2 due to potential resistance pathways. Additionally, the study demonstrates alterations in glutamine and glucose dependencies, glutamine oxidation, and intracellular metabolite levels upon ASCT2 knockout in TNBC cells, providing insights into how cancer cells adapt to metabolic perturbations. Proteomic and mRNA sequencing analyses suggest upregulation of proteins associated with macropinocytosis in ASCT2 knockout cells, further implicating this pathway in resistance mechanisms.

Overall, the study is well conducted, and the conclusions will be relevant to a broad audience. However, there are several points that need to be addressed before publication. Below are my main concerns:

1. In Figure 4, the authors demonstrate increased expression of multiple proteins associated with macropinocytosis, inhibition of ferroptosis, and glutamine efflux in HCC1806 ASCT2 KO cells through whole proteome analysis. However, these results were not validated by a different approach. The expression of selected proteins should be assessed by western blot to solidify these results.

After discussion with the editor, we agreed that validating proteins involved in our macropinocytic phenotype would be most beneficial. We therefore ordered

antibodies to the 2 most significantly upregulated macropinocytosis proteins from Figure 4C, SRC and SNX9. We collected lysates from WT, KO2 and KO3, performing 5 biological repeats. As shown below, we confirmed upregulation of both these proteins in the two knockout lines, supporting our quantitative proteomics data and the 70kDa dextran uptake data.

These new data strengthen the macropinocytosis phenotype, and have been added to the manuscript in Figure 4D, with additional text added on Page 9 as follows:

“We confirmed the increased protein expression of SRC and SNX9 by western blot (Fig. 4D).”

New Figure 4D

2. The authors conclude that macropinocytosis mediates resistance to the loss of glutamine transport and provide evidence that fluorescently labeled 70 kDa dextran signal is increased in ASCT2 KO cells. However, this evidence may be insufficient for a solid conclusion. This reviewer recommends measuring lysosomal proteolysis using DQ BSA, a self-quenched albumin probe whose fluorescence becomes dequenched upon lysosomal proteolysis in ASCT2 KO cells.

3. Along the same lines, if extracellular material can replenish the glutamine pool and sustain cell proliferation after lysosomal fusion and degradation of macropinosomes, the authors should demonstrate that the addition of BSA to medium without glutamine rescues the proliferative defect in ASCT2 KO cells but not in NC cells.

4. Are ASCT2 KO cells more sensitive to inhibitors of lysosomal activity?

For ease, we will answer these three related questions together.

We appreciate the reviewers point that protein breakdown may contribute to the ASCT2 knockout rescue, and have therefore undertaken the suggested experiments which have now been added to the manuscript (see below). However, it is important to note that it has previously been shown in breast cancer cell lines that protein (e.g. BSA) breakdown in the lysosome may not be sufficient for mediating macropinocytosis-mediated rescue of cell growth.

Jayashankar and Edinger (*Nature Communications* 11:1121, 2020) showed in macropinocytotic breast cancer cells that BSA alone is unable to rescue cell growth at low (1%) amino acid levels. As shown below (Figure 1C from their *Nature Comms* paper), at 1% amino acids, BSA alone has little to no effect in the macropinocytotic MCF-7 and T-47D cell lines, whereas necrotic cell debris, which includes amino acids, nucleotides, sugars and fatty acids, can rescue proliferation at low amino acid

levels. By comparison, the non-macropinocytotic cell line HCC1569, which we also used as our control line, could not be rescued by either BSA or necrotic cell debris.

In our manuscript, our data show that the free amino acid content is critical for the rescue of glutamine metabolism in the ASCT2KO cell lines. As shown in Figure 3A (see below) of our manuscript, despite low glutamine transporter activity, the majority of intracellular free glutamine after a 24 h pulse of $^{13}\text{C}_5$ -Glutamine (in both HCC1806 and MDA-MB-231 cells) came from ^{13}C -labelled free glutamine in the media, not from unlabelled ^{12}C -Gln from lysosomal breakdown of proteins brought in by macropinocytosis. The black ^{12}C -portion present in HCC1806 cells was also unlikely to come from protein breakdown, as shown by the $^{13}\text{C}_6$ -Glucose tracing in Figure 3I (also shown below), whereby glucose carbons contribute to free glutamine levels in HCC1806 cells through exit of the TCA cycle at α -KG.

Original Figure 3

These data clearly show that the free amino acid levels (together with free glucose in HCC1806 cells) present in the culture media are sufficient to restore intracellular glutamine levels, supporting glutamine metabolism and cell growth of ASCT2KO cells.

The experiments suggested by Reviewer 1 have further backed up these data, and solidify this mechanism of free amino acid rescue, since BSA alone was indeed unable to rescue the ASCT2KO cell growth, and there was no difference in sensitivity to chloroquine in the ASCT2KO cells compared to control.

We have added two clarifications in the results and discussion to ensure we better highlight the key piece of data from the original manuscript, together with the new data examining the role of BSA breakdown by macropinocytosis:

In the results on page 8, changes underlined:

“Interestingly, despite a significant reduction in the rate of glutamine uptake (Fig. 1F), intracellular $^{13}\text{C}_5$ -labelled glutamine levels were significantly higher in both HCC1806 and MDA-MB-231 A2KO cells (Fig. 3A). In HCC1806 A2KO cells, both ^{13}C -labelled isotopologues (m5 and m3) and the unlabelled glutamine isotopologue (m0) were significantly increased, indicating that in these cells, glutamine supplies are being replenished by another source in addition to extracellular ^{13}C -labelled glutamine such as *de novo* glutamine synthesis or by blocking the efflux of intracellular glutamine.”

In the results on Page 12, new text added:

“Macropinocytosis is a non-specific mechanism mediating uptake of extracellular components, and we next sought to clarify the mechanism by which macropinocytosis rescues the A2KO cells, and whether this process relied on free amino acid uptake, or on protein breakdown. We first attempted to rescue cell growth using BSA as a protein source in glutamine-free conditions, since BSA could be broken down to produce amino acids after fusion of the macropinosome to lysosomes. Similar to previous published data (Jayashankar and Edinger 2020), free protein (BSA) was unable to rescue HCC1806KO cell growth in the absence of free glutamine (Fig. EV5B). Using de-quenched BSA (DQ-BSA) and lysotracker colocalisation analysis, we confirmed that macropinosome/lysosome fusion was occurring, however there were minimal changes in colocalisation between WT and KO cells (Fig. EV5C). Similarly, there was no difference in WT or KO cells treated with the lysosomal inhibitor chloroquine (Fig. EV5D), further confirming that protein breakdown was not a substantial component of the KO cell resistance phenotype. By comparison, our ^{13}C -glutamine tracing data (Fig 3A) clearly suggest that extracellular free glutamine, and to some extent free glucose, are the critical macropinocytosed nutrients being utilised by HCC1806 A2KO cells.”

New Figure EV5

In the discussion on Page 15/16, new text added:

“Previous data have shown that breast cancer cells require more than just macropinocytic protein (e.g. BSA) to support growth (Jayashankar and Edinger 2020), and it was clear that our A2KO cells similarly replenished glutamine from ¹³C-labelled glutamine or glucose (Fig. 3A), and not from BSA (Fig. EV5D), nor were A2KO cells more sensitive to chloroquine inhibition (Fig. EV5F).”

Referee #2 comments:

In the manuscript under review the authors investigate the dependency of triple-negative breast cancer (TNBC) on the plasma membrane glutamine transporter ASCT2 for glutamine uptake to fuel their metabolism and growth. This study is based on their previous identification of ASCT2 as highly upregulated in TNBC. They find that breast cancer cell lines tolerate complete loss of ASCT2 surprisingly well, and deletion of ASCT2 did not result in a decrease in intracellular glutamine levels, and instead were higher than in wild-type cells. The authors trace these metabolic changes back to an upregulation of micropinocytosis, reduction of glutamine efflux through downregulation of SLC7A5, and increased glutamine synthesis in ASCT2 knockout cells, implicating these as a possible resistance mechanisms to inhibiting glutamine uptake through ASCT2 in the context of breast cancer chemotherapy.

By testing whether targeting glutamine uptake by inhibition of ASCT2 could be a viable strategy in the treatment of triple negative breast cancers, the paper seeks to identify new strategies to treat triple negative breast cancers, which in principle is a question of high clinic relevance.

The approach the authors take to directly target ASCT2 is limited in scope but yields some novel insights into the regulation of glutamine homeostasis in breast cancer.

In general, the conclusions of the study are supported by the experiments. However, some major concerns need to be addressed that are essential to support the main conclusions of the study.

1 - Most of the findings rely on ASCT2 KO cells generated by CRISPR/Cas, and the authors target two different exons. They validate loss of ASCT2 protein by western blot.

- As the authors write in the discussion, it is possible that targeting of ASCT2 with a guide targeting exon 7 (and 4 of that matter) results in partial loss of the protein. Thus, it is critical to validate the KO cells appropriately. Next generation sequencing or another sequencing method should be employed to confirm the mutations introduced to the proposed KO cells and determine whether and what part of the coding sequence of ASCT2 is actually deleted in the different clones.

We thank the reviewer for pointing out this perceived lack of validation. As mentioned by the reviewer, we validated the knockout using 2 different CRISPR guides as well as western blot – however more importantly we also functionally characterised the knockout using the [³H]-L-glutamine uptake assay, showing that there was a significant loss of glutamine uptake in all KO clones. Even if a truncated protein was

expressed, losing the additional transmembrane regions present in the c-terminal regions would be unlikely result in a functional glutamine transporter.

To address the reviewer's questions, we have now included further validation data that was not presented in our original submission (new Appendix Figure S1). These data show the single insertion of an A (shown in green; S1A) in HCC1806 cells results in a frameshift in ASCT2, and the insertion of 23 bases in MDA-MB231 (shown in green; S1A), also resulting in a frame shift. These frameshifts would result in a premature stop codon leading to a truncated ASCT2 protein.

We also analysed our proteomics data, which detected two peptides that were upstream of both the exon 7 and exon 4 guides in WT cells (S1C yellow highlight). If there was a truncated protein, these would be present in the KO lines, however neither of these peptides were expressed in A2KO cells.

We have added additional text on Page 14 of the manuscript to ensure this is clear and highlight the new Appendix Figure S1, new text is underlined:

“One explanation for the lack of growth inhibition could be that the mutation caused by exon 7 CRISPR guide results in a partially functional protein or isoform that is not detectable by ASCT2 antibodies. We mapped the frameshift mutations (mRNAseq and Sanger sequencing) and used our proteomics data to show there were no ASCT2 peptides in KO cells upstream of the guide RNA (Appendix Figure S1). In addition, since targeting exon 4 also resulted in the same phenotype, this suggests that it is unlikely that a partially functional ASCT2 may be responsible for the growth phenotype observed in A2KO cells (Fig. EV1N-Q).”

New Appendix Figure S1

B

Cell lines	Sequence of SLC1A5 Exon 7 gRNA site with insertion
HCC1806 NC	GGCCATCATCCTCGAAGCAGTCAACCTCCCGTCGACCATATCTCCTT
HCC1806 A2KO	GGCCATCATCCTCGAAAGCAGTCAACCTCCCGTCGACCATATCTCCTT
MDA-MB-231 NC	GGCCATCATCCTCGAAGCAGTCAACCTCCCGTCGACCATATCTCCTTGATCCT
MDA-MB-231 A2KO	GGCCATCATCCTCGAAATGCGCGGCTCCAGGACCCCTGATCCTCCTTGCTGT

C

MVADPPRDSKGLAAEPTANGGLALASIEDQGAAGGYCGSRDQVRRCLRANLLVLLTVVAVGVALGL
 GYSGAGGALALGPERLSAFVFPGELELLRLLRMILPLVVCSLIGGAASLDPGALGRLGAWALLFFLVTTLLAS
 ALGVGLALALQPGAASAAINASVGAAGSAENAPSK**EVLDSFLDLARNIFPSNLVSAAFRS**YSTTYEERNITG
 TRVKVPVGGVEYEGMNILGLVVFVAVFGVALRKLGPGEELLIRFFNSFNEATMVLVSWIMWYAPVGMFLVAG
 KIVEMEDVGLLFARLKGKYLCCLLGHAIHGLLVPLIYFLFTRKNPYRFLWGIVTPLATAFGTSSSSATLPLMM
KCVEENNGVAKHISRIFLPIGATVNMMDGAALFQCVAAVFIAQLSQSLDFVKIITILV**TATASSVGAAGIPAGGV**
 LTLAIIIEAVNLPVDHISLILAVDWLVDRSCTVLNVEGDALGAGLLQNYVDRTERSRSTPELIQVK**SELPLDPL**
 PVPTEEGNPLLKHYRGPAGDATVASEKESVM

ASCT2 peptides detected in Proteomics
 Exon 4 coded amino acids
 Exon 7 coded amino acids

2 - Because of potential off-target effects of CRISPR, some of the key experiments (such as the proliferation and uptake assays in Figure 1, the macropinocytosis assays in Figure 5 and possibly xenograft studies) should include control cells in which the (guide refractory) cDNA for ASCT2 is added back to the KO cells. While this is an important control to determine whether observed phenotypes are specific to ASCT2 loss, they might also reveal how flexible the metabolic rewiring in the breast cancer cells is, i.e. whether reexpression of ASCT2 can revert back all phenotypes or whether cells undergo a change that is irreversible or overall metabolically more favorable, which could be essential information for developing ways to target glutamine metabolism in cancer.

This was an intriguing question. Fortunately, we had already undertaken an overexpression experiment for a related project, but had not thought to undertake the experiments requested. We therefore revived our frozen cell stocks and were somewhat surprised to see that indeed, the increased macropinocytosis is a reversible process, with ASCT2 re-expression/rescue decreasing macropinocytosis.

We have included new data for this overexpression cell line including a western blot for protein expression, [³H]-L-Glutamine uptake assay, proliferation data and the 70 kDa dextran macropinocytosis assay.

These data have now been added to EV5, with new text on Page 12 as follows:

“We next determined whether ASCT2 re-expression could revert the macropinocytic phenotype in A2KO cells. Overexpression of ASCT2 restored protein expression (Fig. EV5E) and glutamine uptake (Fig. EV5F) in A2KO cells, and while this did not change proliferative capacity (Fig. EV5G), it reduced macropinocytosis levels back to control (Fig. EV5H), suggesting that intracellular glutamine levels drive this adaptive yet reversible process.”

New Figure EV5

Dear Dr Holst, dear Dr Wahi,

Thank you for submitting your revised manuscript (EMBOJ-2024-117102R) to The EMBO Journal. Your amended study was sent back to the two referees for their re-evaluation, and we have received comments from both of them, which I enclose below. As you will see, the experts stated that the work has been substantially improved by the revisions and they are now broadly in favour of publication.

Thus, we are pleased to inform you that your manuscript has been accepted in principle for publication in The EMBO Journal.

We now need you to take care of a number of minor issues related to formatting and data presentation as detailed below, which should be addressed at re-submission.

Please contact me at any time if you have additional questions related to below points.

As you might have seen on our web page, every paper at the EMBO Journal now includes a 'Synopsis', displayed on the html and freely accessible to all readers. The synopsis includes a 'model' figure as well as 2-5 one-short-sentence bullet points that summarize the article. I would appreciate if you could provide this figure and the bullet points.

Thank you for giving us the chance to consider your manuscript for The EMBO Journal.
I look forward to your final revision.

Again, please contact me at any time if you need any help or have further questions.

Best regards,

Daniel Klimmeck

>> Please add up to five keywords to your study.

>> Author Contributions: Please remove the author contributions information from the manuscript text. Note that CRediT has replaced the traditional author contributions section as of now because it offers a systematic machine-readable author contributions format that allows for more effective research assessment. and use the free text boxes beneath each contributing author's name to add specific details on the author's contribution.

More information is available in our guide to authors.
<https://www.embopress.org/page/journal/14602075/authorguide>

>> Callouts: "Fig. 1R" and "Fig. 1S" should be corrected to the correct callout; Please recheck reference to Table 1 (is it table on p. 17-18?) In that case we need a legend with title - Table 1 - above the table; Table 1 would then need to be placed between main and EV figure legends.

>> Dataset EV legends: Table EV1 (Proteomics data table) and Table EV2 (RNA-seq data table) are datasets and need to be updated as such, manuscript callouts included; correct nomenclature is Dataset EV1, Dataset EV2

>> Appendix file: the legend for the Appendix Figure S1 should be removed from the manuscript and provided in a PDF file together with the Appendix Figure S1; the PDF is called 'Appendix' and needs to have a short ToC on the title page and the

page number to show where the figure is located in the file.

>> References: please adjust the reference format to EMBO Journal format, 10 authors et al, and place References after the Discussion, before figure legends.

>> Data availability section: please provide hyperlinks for the GEO and PRIDE datasets. Remove the referee tokens and make sure dataset privacy is released.

>> Adjust the 'Materials & Methods' section to 'Methods'.

>> Add a Reagents and Tools table to the Methods section, listing key reagents, experimental models, software and relevant equipment.

>> Remove the 'data not shown' statement on p.23 ("Error bars are not shown..."), or add respective information.

>> Please remove the 'Paper Explained' section from the manuscript.

>> Recheck the bioRxiv citation Lambies et al (2024) for journal publication and update the journal reference in case.

>> Figure legends need to be rearranged: main figure legends should all be listed first and then all EV figure legends should follow

>> Consider additional changes and comments from our production team as indicated below:

Figure Legends - Comments

- Please define the annotated p values **** as well as provide the exact p-values for the same in the legend of figure EV 1p; as appropriate.
- Please note that the exact p values are not provided in the legends of figures 1f, m, q, s; 2a, c, e, g, i, l; 3a-i; 5a, c; 6b-d; EV 1c, e, g, k; EV 2a, c; EV 3b-e; EV 4d-f; EV 5c-d, h.
- Please indicate the statistical test used for data analysis in the legends of figures 4b; EV 1p; EV 4a-b, g.
- Please note that in figures 1f, h-i, m-o, q-s; 5a, c-e; 6b-d; EV 2a, c; EV 5d, f; there is a mismatch between the annotated p values in the figure legend and the annotated p values in the figure file that should be corrected.
- Please note that for the figure 1g, p-values and statistical tests are indicated in the legends. However, comparison for the same, "****/**/**/*" has not been represented in the figure. Please rectify this in the figure or legend as applicable.
- Please note that information related to n is missing in the legends of figures 3a-i; EV 1p-q; EV 3b-e.
- Although 'n' is provided, please describe the nature of entity for 'n' in the legend of figure 4d.
- Please note that the error bars are not defined in the legends of figures 1m, q; 3a-i; 4d; EV 1p-q; EV 3b-e.
- Please note that for heatmap present in figures 4a, c, e-f; EV 4a-b, g; a numbered scale bar is not provided. This needs to be rectified.
- Please note that the scale bar needs to be defined for figures 5a, c-e; 6e; EV 5c, h.

Please use the link below to submit your revision:

Referee #1:

The revised version by Wahi K. and colleagues provides additional experiments and validations of key findings, which enhance

the robustness of the study. The detailed responses and inclusion of new data satisfactorily address all the concerns of this reviewer. I recommend publication in EMBO Journal.

Referee #2:

The authors have addressed all comments sufficiently and I have no more concerns about the manuscript being published in its current form.

Comments from the editor:

As you might have seen on our web page, every paper at the EMBO Journal now includes a 'Synopsis', displayed on the html and freely accessible to all readers. The synopsis includes a 'model' figure as well as 2-5 one-short-sentence bullet points that summarize the article. I would appreciate if you could provide this figure and the bullet points.

Model figure and synopsis have been added and uploaded.

>> Please add up to five keywords to your study.

Keywords added to manuscript title page.

>> Author Contributions: Please remove the author contributions information from the manuscript text. Note that CRediT has replaced the traditional author contributions section as of now because it offers a systematic machine-readable author contributions format that allows for more effective research assessment. and use the free text boxes beneath each contributing author's name to add specific details on the author's contribution.

Author contributions removed from the manuscript and added to each author in the online system.

>> Callouts: "Fig. 1R" and "Fig. 1S" should be corrected to the correct callout;

Text has been edited to clarify.

Please recheck reference to Table 1 (is it table on p. 17-18?) In that case we need a legend with title - Table 1 - above the table; Table 1 would then need to be placed between main and EV figure legends.

Moved Table 1 and added a legend as requested.

>> Dataset EV legends: Table EV1 (Proteomics data table) and Table EV2 (RNA-seq data table) are datasets and need to be updated as such, manuscript callouts included; correct nomenclature is Dataset EV1, Dataset EV2.

Changed to Dataset EV in callouts and upload, legends added to end of manuscript file.

>> Appendix file: the legend for the Appendix Figure S1 should be removed from the manuscript and provided in a PDF file together with the Appendix Figure S1; the PDF is called 'Appendix' and needs to have a short ToC on the title page and the page number to show where the figure is located in the file.

Appendix file has been added and uploaded as requested.

>> References: please adjust the reference format to EMBO Journal format, 10 authors et al, and place References after the Discussion, before figure legends.

Format changed as requested.

>> Data availability section: please provide hyperlinks for the GEO and PRIDE datasets. Remove the referee tokens and make sure dataset privacy is released.

Referee tokens removed and links to final datasets added. GEO dataset privacy released, awaiting final release of PRIDE and will notify once it is confirmed.

>> Adjust the 'Materials & Methods' section to 'Methods'.

Done.

>> Add a Reagents and Tools table to the Methods section, listing key reagents, experimental models, software and relevant equipment.

Done.

>> Remove the 'data not shown' statement on p.23 ("Error bars are not shown..."), or add respective information.

Done.

>> Please remove the 'Paper Explained' section from the manuscript.

Done.

>> Recheck the bioRxiv citation Lambies et al (2024) for journal publication and update the journal reference in case.

This is still in bioRxiv as at the 16th September 2024.

>> Figure legends need to be rearranged: main figure legends should all be listed first and then all EV figure legends should follow.

Done.

>> Consider additional changes and comments from our production team as indicated below:

Figure Legends - Comments

- Please define the annotated p values **** as well as provide the exact p-values for the same in the legend of figure EV 1p; as appropriate.

Done.

- Please note that the exact p values are not provided in the legends of figures 1f, m, q, s; 2a, c, e, g, i, l; 3a-i; 5a, c; 6b-d; EV 1c, e, g, k; EV 2a, c; EV 3b-e; EV 4d-f; EV 5c-d, h.

Done.

- Please indicate the statistical test used for data analysis in the legends of figures 4b; EV 1p; EV 4a-b, g.

Done.

- Please note that in figures 1f, h-i, m-o, q-s; 5a, c-e; 6b-d; EV 2a, c; EV 5d, f; there is a mismatch between the annotated p values in the figure legend and the annotated p values in the figure file that should be corrected.

Done.

- Please note that for the figure 1g, p-values and statistical tests are indicated in the legends. However, comparison for the same, "****/**/*/*" has not been represented in the figure. Please rectify this in the figure or legend as applicable.

Done.

- Please note that information related to n is missing in the legends of figures 3a-i; EV 1p-q; EV 3b-e.

Done.

- Although 'n' is provided, please describe the nature of entity for 'n' in the legend of figure 4d. Done.

- Please note that the error bars are not defined in the legends of figures 1m, q; 3a-i; 4d; EV 1p-q; EV 3b-e.

Done.

- Please note that for heatmap present in figures 4a, c, e-f; EV 4a-b, g; a numbered scale bar is not provided. This needs to be rectified.

New text added to clarify, as a numbered scale bar is not appropriate for this analysis:

“A relative colour scheme is used for heatmaps (A, C, E, F) where minimum (blue) and maximum (red) values in each row are converted to colours (<https://software.broadinstitute.org/morpheus/>).”

- Please note that the scale bar needs to be defined for figures 5a, c-e; 6e; EV 5c, h.

Done.

Dear Dr Holst,

Thank you for submitting the revised version of your manuscript. I have now evaluated your amended manuscript and concluded that the remaining minor concerns have been sufficiently addressed.

I am thus pleased to inform you that your manuscript has been accepted for publication in the EMBO Journal.

On a different note, I would like to alert you that EMBO Press offers a format for a video-synopsis of work published with us, which essentially is a short, author-generated film explaining the core findings in hand drawings, and, as we believe, can be very useful to increase visibility of the work. Please see the following link for representative examples and their integration into the article web page:

<https://www.embopress.org/doi/full/10.15252/emj.2019103932>

Finally, we have noted that the submitted version of your article is also posted on the preprint platform bioRxiv. We would appreciate if you could alert bioRxiv on the acceptance of this manuscript at The EMBO Journal in order to allow for an update of the entry status. Thank you in advance!

Best regards,

Daniel Klimmeck

Daniel Klimmeck, PhD
Senior Editor
The EMBO Journal
EMBO
Postfach 1022-40
Meyerohofstrasse 1
D-69117 Heidelberg

contact@embojournal.org
Submit at: <http://emboj.msubmit.net>
